# Observation based temperature and freshwater noise over the Atlantic Ocean

Amber A. Boot[1] and Henk A. Dijkstra[1,2]

[1]Institute for Marine and Atmospheric research Utrecht, Department of Physics, Utrecht University, Utrecht, the Netherlands
[2]Center for Complex Systems Studies, Utrecht University, Utrecht, the Netherlands

**Correspondence:** Amber A. Boot (a.a.boot@uu.nl)

**Abstract.** The ocean is forced at the surface by a heat flux and freshwater flux field from the atmosphere. Short time-scale variability in these fluxes, i.e. noise, can influence long-term ocean variability and might even affect the Atlantic Meridional Overturning Circulation (AMOC). Often this noise is assumed to be Gaussian, but detailed analyses of its statistics appear to be lacking. Here we study the noise characteristics in reanalysis data for two fields which are commonly used to force ocean-only models: evaporation minus precipitation and 2 m air temperature. We construct several noise models for both fields, and a point-wise Normal Inverse Gaussian distribution model gives the best performance. An analysis of CMIP6 models shows that these models do a reasonable job in representing the standard deviation and skewness of the noise, but the excess kurtosis is more difficult to capture. The point-wise noise model performs better than the CMIP6 models and can be used as forcing in ocean-only models to study, for example, noise-induced transitions of the AMOC.

## 1 Introduction

The ocean is forced at the surface by momentum, heat and freshwater fluxes from the atmosphere. Since the ocean responds relatively slowly to the atmospheric forcing, anomalies in this forcing can be modelled as a noise process (Hasselmann, 1976). This study is motivated by the role of such noise in causing noise-induced transitions in the Atlantic Meridional Overturning Circulation (AMOC). The AMOC has a major influence on global, and specifically, Northern Hemispheric climate and has been identified as one of the potential major tipping points in the Earth System (Lenton et al., 2008; McKay et al., 2022). A collapse or strong weakening of the AMOC has major consequences for the climate system by changing e.g. global temperature patterns (van Westen et al., 2024b), atmospheric circulation (Orihuela-Pinto et al., 2022), Arctic sea ice cover (van Westen et al., 2024b), the global carbon cycle (Zickfeld et al., 2008; Boot et al., 2024b) and marine ecosystems (Schmittner, 2005; Boot et al., 2024a).

Simple box models have shown that the AMOC can show noise-induced transitions (Castellana et al., 2019; van Westen et al., 2024a) and probabilities of such transitions could be obtained using rare-event techniques. In these types of studies, the noise is applied only in the freshwater flux and is often assumed to be white for simplicity. Recently, noise induced transitions have also been studied in an Intermediate Complexity Earth System Model (EMIC; Cini et al., 2024) using rare event techniques. Ideally, one would want to study the transitions in full complexity, CMIP6-type, Earth System Models (ESMs). However, due

to the complexity and cost of these models, it is not yet possible to systematically use these ESMs for these types of studies. Recently, a study did look at AMOC tipping in a 10-member ensemble of the NASA-GISS ESM, showing that under the same forcing some ensemble members simulate an AMOC recovery under future emissions, and others show a consistent weakening (Romanou et al., 2023). However, the AMOC does not show a complete collapse in these ensemble members.

To determine the probability of noise-induced transitions using rare event techniques one is at the moment restricted to using ocean-only models and hence the specification of the atmospheric noise is crucial. However, to our knowledge, a detailed study on the properties of the noise in the actual fields relevant in the forcing of ocean models is lacking. Here we focus on noise in the freshwater flux ($E - P$) and in the 2 m air temperature ($T_{2m}$). Noise in the momentum flux related to surface winds might also be important for the AMOC. However, we do not consider this here for two main reasons: the statistical properties of the surface winds have been studied more thoroughly before (Sura, 2003; Monahan, 2004, 2018), and the noise in the momentum flux is less important for simulating noise-induced transitions of the AMOC.

Such a study is also useful to determine whether EMICs and ESMs adequately capture these noise fields. We know that these types of models exhibit, sometimes very large, biases in their mean state, but also in variability on a whole range of timescales. For example, $T_{2m}$ is biased too warm in the CMIP6 models over the Atlantic sector of the Southern Ocean and the Eastern South Atlantic, while there is a cold bias over much of the North Atlantic and Arctic Ocean (IPCC AR6 Chapter 3). The air temperature biases can also be seen in the sea surface temperatures (Zhang et al., 2023), thereby directly affecting the density structure of the ocean. For precipitation there is a consistent double Intertropical Convergence Zone (ITCZ) bias from CMIP3 to CMIP6 models (Tian and Dong, 2020). This means that in the Atlantic, the ITCZ, and therefore bands of high precipitation extend too much towards the south. Following the double ITCZ bias (Tian and Dong, 2020; Li et al., 2020), there is a strong positive freshwater flux bias north of the equator and a strong negative bias south of the equator in the CMIP6 multi model mean (MMM; Liu et al., 2022). Between 10° and 60°N, and the equator and 35°S the freshwater flux is typically positively biased in the CMIP6 MMM (Liu et al., 2022). These biases are among the reasons why the AMOC is thought to be too stable in CMIP6 type models (Weijer et al., 2019; van Westen and Dijkstra, 2024).

In this study we determine the statistical properties of the $E - P$ and $T_{2m}$ noise based on the ERA5 reanalysis data. We compare this observation-based noise with the noise simulated by coupled CMIP6 ESMs and identify relevant biases. Based on the ERA5 noise we construct a noise model that can be used to force ocean-only models. This product can be used to study the influence of short timescale atmospheric variability on long term ocean variability and eventually to study noise-induced transitions of the AMOC.

## 2 Methods

### 2.1 ERA5 reanalysis data

We analyse the noise in $E - P$ and $T_{2m}$ over the Atlantic Ocean between 60°S and 80°N. For this we use ERA5 reanalysis data (Hersbach et al., 2020), which is the most recent reanalysis product of the European Center for Medium-Range Weather Forecasts (ECMWF) and replaces the ERA-Interim reanalysis product. The ERA5 product is created by combining both satel-

lite and ground observations with a numerical model used for weather forecasting. For the freshwater flux we determine the net freshwater forcing by taking the sum of the variables 'total precipitation' and 'evaporation', i.e. - (total precipitation + evaporation). Since evaporation is defined negative in ERA5 data, and total precipitation positive, this results in a dataset for $E - P$ where net evaporation is positive, and net precipitation is negative. The datasets contain monthly data from 1940 to 2022 on a $0.25°$ rectilinear grid. To determine the noise in both fluxes, we first detrend each grid point by subtracting a 5-year running mean. Next, we deseasonalise the data by subtracting a monthly climatology based on the detrended data. This results in a noisy dataset where each grid point has zero mean and no trend. We analyze the fields by looking at the standard deviation ($\sigma$), skewness and excess kurtosis of the noise, where Gaussian white noise would have zero skewness and zero excess kurtosis.

## 2.2 CMIP6 models

We compare the noise in the ERA5 data to that found in CMIP6 ESMs. In total we use 36 different models, where we note that we use two different realizations from the UKESM-1-0-LL model that is run by two different model groups (i.e. the Met Office Headly Centre (MOHC) and National Institute of Meteorological Sciences - Korea Meteorological Administration (NIMS-KMA)). For each model we determine the evaporation minus precipitation by using the variables 'evspsbl' and 'pr', and we use 'tas' for $T_{2m}$. We do this for the historical simulations between 1940 and 2014. We first regrid all models to a $1° \times 1°$ rectilinear grid. Next, we compute the noise in the models by following the same methodology as for the ERA5 data, i.e. we detrend and deseasonalise the data. We could also use simulations without forcing, i.e. the piControl simulations. However, for comparison with the ERA5 data it is better to use a similar methodology for both data sources, which includes the detrending in the ERA5 data.

When comparing the CMIP6 data to the ERA5 data we use the same time period in the ERA5 data as in the CMIP6 data, i.e. 1940 to 2014, and re-grid the ERA5 data to a $1°$ rectilinear grid. Due to its original higher resolution, the land mask in ERA5 captures small islands that are not captured by the CMIP6 land mask. To account for this, we mask out these small islands in both the ERA5 and the CMIP6 data. A full list of the models used, including which member and citations can be found in the Appendix (Table A1).

## 3 Results

### 3.1 ERA5 reanalysis data

Fig. 1 (a-c) shows the standard deviation, skewness and excess kurtosis in the $E - P$ noise. The highest standard deviation is found north of the equator in the ITCZ, and other regions with relatively high standard deviations are the western boundary currents. In the Northern Hemisphere, the strongest negative skewness is found between $10°N$ and $30°N$ (Fig. 1b). The negative skewness here indicates that the distribution is skewed towards extreme precipitation events, which is partially related to tropical storm activity in this region. In the Southern Hemisphere there is a strong negative skewness in the region of the South Equatorial Current. South of there, there is a small region of moderate positive skewness. The rest of the ocean generally shows

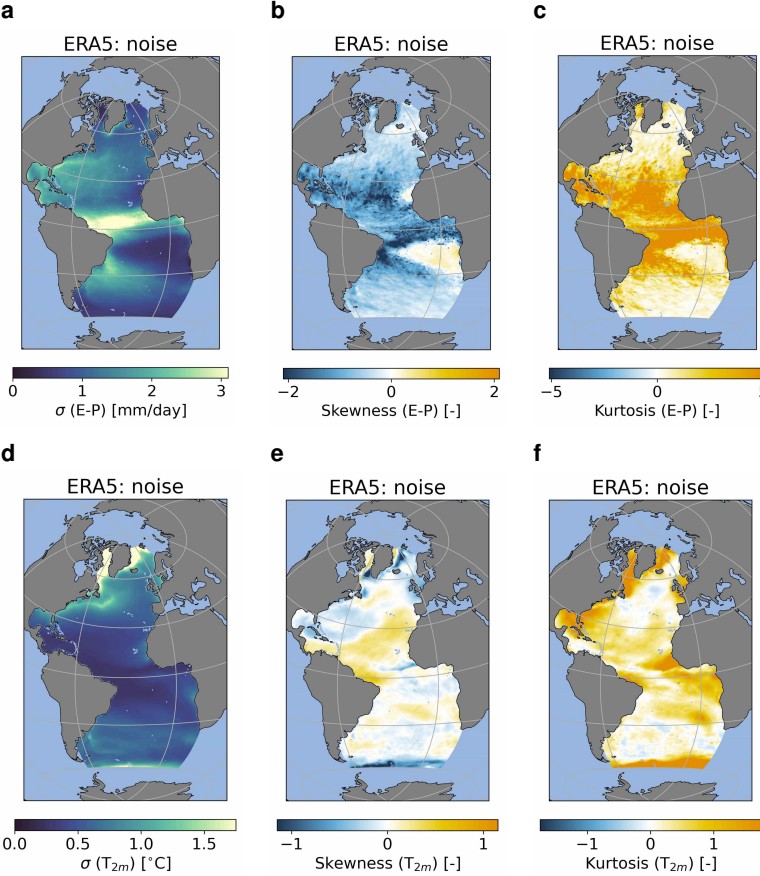

**Figure 1.** Standard deviation ($\sigma$), skewness and excess kurtosis over time for the ERA5 noise for the E − P flux (a-c) and $T_{2m}$ (d-f).

a small negative or near zero skewness. The excess kurtosis shows a relatively similar pattern as the skewness except with the opposite sign (Fig. 1c). The strongest positive excess kurtosis is found over the entire latitudinal band 10°S to 30°N. Also this is an indication of high extremes, and because of the negative skewness it indicates extreme precipitation events. The rest of the ocean has slightly positive or near zero excess kurtosis. Due to the non-zero skewness and excess kurtosis in the noise in most grid points, the noise cannot be classified as Gaussian white noise in these grid points.

For $T_{2m}$ (Fig. 1d-f), the largest standard deviation in the noise is found in the (seasonally) sea ice covered regions in the high latitude North Atlantic (Fig. 1d). Also the Gulfstream region shows a relatively high standard deviation. Regions around the sea-ice edge, both in the Northern and Southern Hemispheres, show a relatively strong negative skewness (Fig. 1e), which means the distribution in these regions are skewed towards more cooling events. The pattern for the skewness in the South Atlantic is relatively patchy with both small negative and small positive values. In the North Atlantic, the regions around the trade winds show positive skewness, and the subtropical gyre shows negative skewness. The (seasonally) sea ice covered

regions show strong negative skewness. For the excess kurtosis (Fig. 1f) most of the Atlantic region shows (strong) positive values with the strongest signals over the sea ice covered regions and close to the seasonal sea-ice edge (also in the South Atlantic), and in the Gulf of Mexico. The combination of negative skewness and positive excess kurtosis in the sea ice covered regions suggests that in these regions strong cooling events can take place which is likely associated with strong increases in sea-ice cover. Just as for the freshwater flux, the excess kurtosis deviates from zero in most regions in the ocean, which means that also the noise in $T_{2m}$ is unlikely to be Gaussian white noise in most grid points.

To better understand the results, we look in the noise fields for regions with similar distributions. We do this by dividing both the $E - P$ and $T_{2m}$ noise fields into 12 different clusters (Fig. 2). Thereto we use a k-means clustering algorithm where we use the standardized standard deviation, skewness and excess kurtosis as input. The decision for 12 clusters is based on several methods (Fig. A1), i.e. the elbow method, the silhouette score, the gap statistic and visually inspecting the clusters while the number of clusters is varied. The probability density functions, standard deviation, skewness and excess kurtosis of the clusters are displayed in Fig. A2 - A5

For $E - P$ we find several relatively large clusters. The subpolar regions are divided into two clusters (clusters 1 and 4) where cluster 4 is more poleward. The main difference between the two clusters is the lower standard deviation in the higher latitude cluster. The high standard deviation region of the ITCZ also clearly stands out as a separate cluster (cluster 6). The subtropical region is divided into 9 different clusters. The 6 clusters that cover the North and South Equatorial Current stand out with strong positive excess kurtosis and strong negative skewness (clusters 2, 3, 5, 9, 10 and 11). These are regions that experience tropical storms and hurricanes, which are recorded as very strong extreme precipitation events in the $E - P$ noise fields. These extremes can be very local, explaining why 6 clusters are necessary for this region. The cluster closest to a Gaussian distribution is the cluster in the Southeastern subtropical region (cluster 12) with a skewness of 0.16 and an excess kurtosis of 0.90 (Fig. 2b). For all clusters, kurtosis is larger than 1.5 times the square of the skewness (Fig. 2b) which is consistent with multiplicative noise (Sardeshmukh and Sura, 2009).

The 12 clusters for the $T_{2m}$ noise do not show an overlap with the $E - P$ clusters and several of them appear to follow the general ocean circulation pattern. For example, cluster 12 is centered around the North Atlantic Current, and cluster 10 around the North Equatorial Current and the North Brazil Current. While for the $E - P$ noise several clusters are necessary for the subtropics, for $T_{2m}$ several clusters are necessary for regions covered by sea ice, or adjacent to these regions. Cluster 3 describes the regions in the Labrador and Greenland Seas that experience sea ice annually. Clusters 2, 5, and 6 all cover regions close to the sea-ice edge. The noise in these regions is likely affected by interannual variability in the sea-ice extent which leads to relatively strong positive excess kurtosis and relatively strong negative skewness. Two clusters (7 and 12) have near zero area weighted skewness and excess kurtosis and are therefore close to a Gaussian distribution (0.00 skewness for both, and 0.06 and 0.13 for excess kurtosis, respectively; Fig. 2d). Cluster 7 covers the South Atlantic between 30°S and 50°S, and parts of the Eastern North Atlantic between 30°N and 60°. Cluster 12 is the cluster around the North Atlantic Current, but does show some variability in both skewness and excess kurtosis in the cluster. Just as for the $E - P$ clusters, for all $T_{2m}$ clusters, kurtosis is larger than 1.5 times the square of the skewness (Fig. 2d) consistent with multiplicative noise.

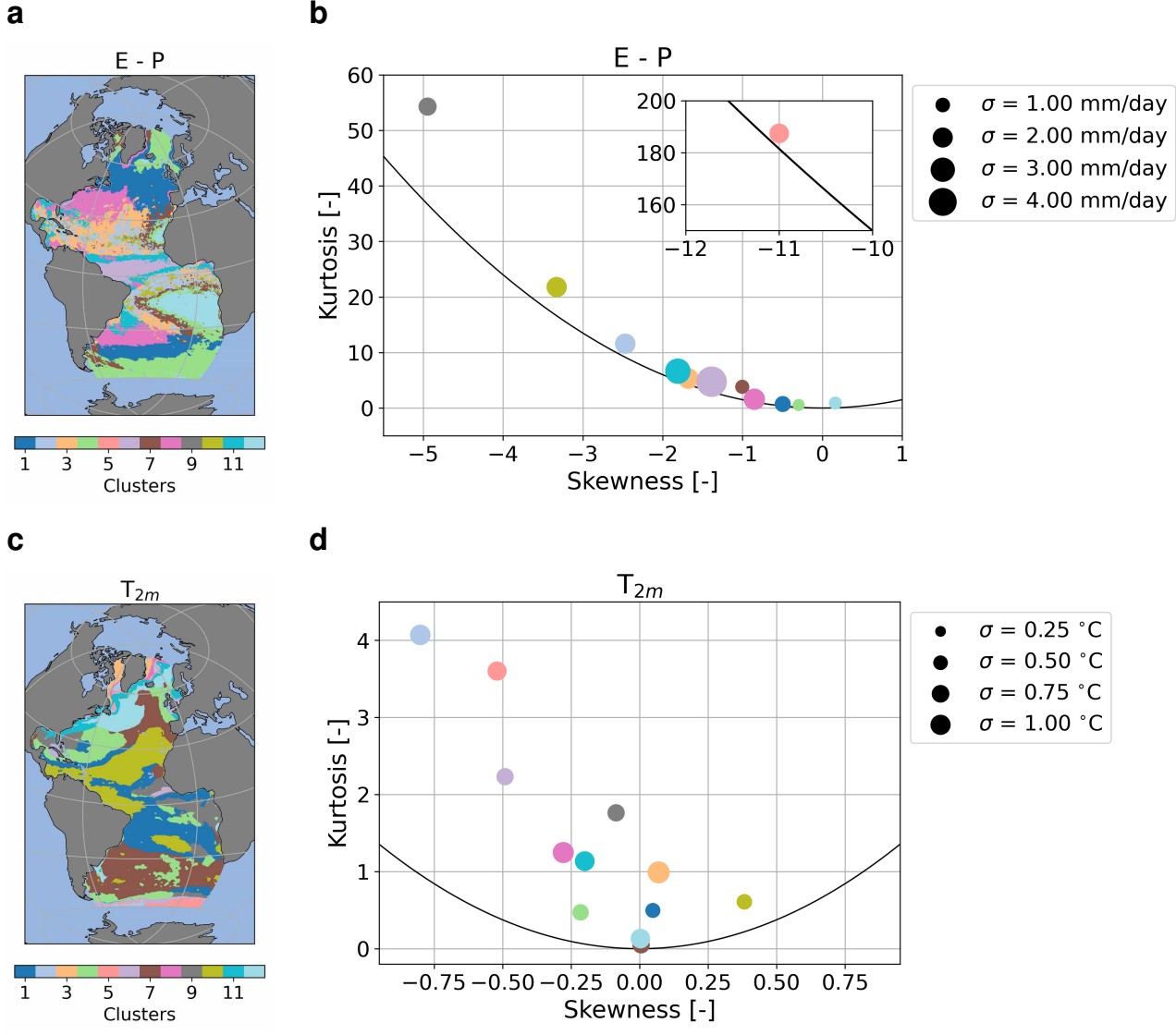

**Figure 2.** Overview of the clusters and corresponding statistics. (a) The clusters for the $E - P$ noise. (b) The skewness (S; x-axis) and excess kurtosis (K; y-axis) of the clusters in (a). The colors of the markers correspond to the color coding in (a). The size of ther markers represents the standard deviation in mm/day. (c) and (d) as in (a) and (b) but for the $T_{2m}$ clusters. The unit of standard deviation in (d) is °C. The black line in (b) and (d) represents K = 1.5S$^2$.

## 3.2 CMIP6 data

In this section we analyze the results for the multi-model mean (MMM) of the CMIP6 models. We determine the MMM at the end of the analysis. This means that we for example first determine the skewness for each model, and then average over the

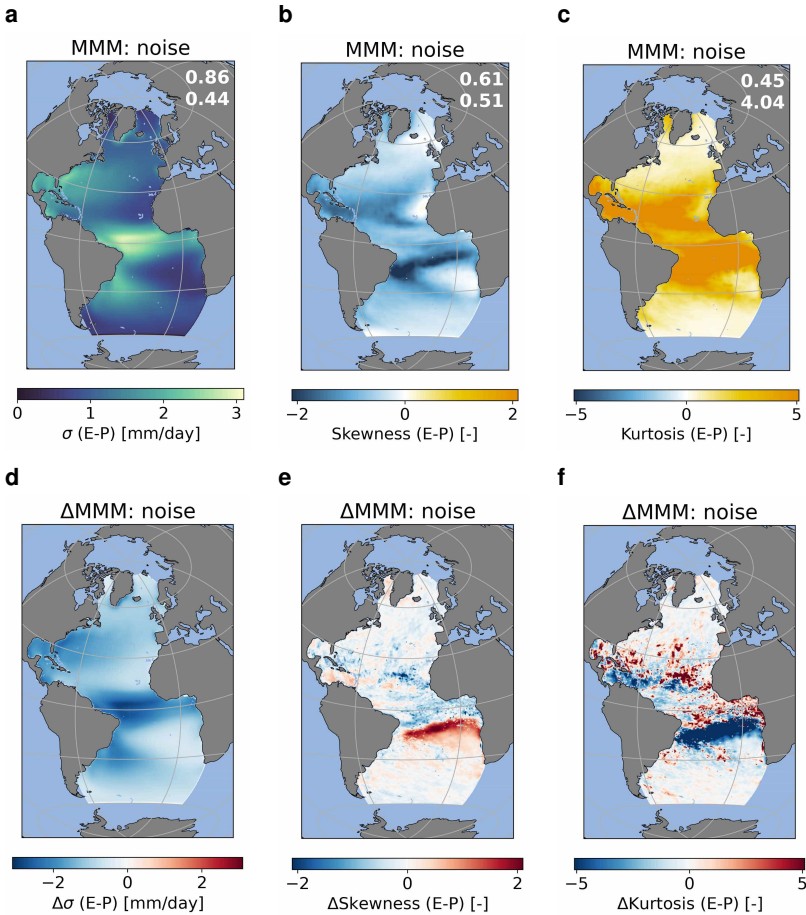

**Figure 3.** Standard deviation ($\sigma$), skewness and excess kurtosis of CMIP6 multi-model mean (MMM) noise for the $E - P$ flux (a) – (c). Differences with ERA5 data (i.e. ERA5 minus CMIP6 MMM) are shown in (d) – (f). The numbers in the top right corner of (a) – (c) reflect the spatial correlation and root mean square error. Units for (a) and (d) are mm/day.

2D skewness fields of all the models to create the MMM. Each model has been given the same weight. Results for individual models can be found in the Appendix (Fig. A10 to Fig. A15).

The MMM for the noise in the $E - P$ flux does not always represent the amplitude in the statistics of the ERA5 noise well (Fig. 3a-f), though the spatial patterns are relatively well resolved in the MMM. The standard deviation is underestimated over the entire ocean with the strongest underestimation in the ITCZ regions and over the western boundary currents (Fig. 3d). The multimodel mean shows a stronger negative skewness over the South Equatorial current that is also shifted more southward compared to ERA5 noise (Fig. 3e). Furthermore, the positive skewness over the eastern subtropical region is not captured by the CMIP6 MMM. The excess kurtosis is also positively biased in the CMIP6 MMM over the South Equatorial Current (Fig.

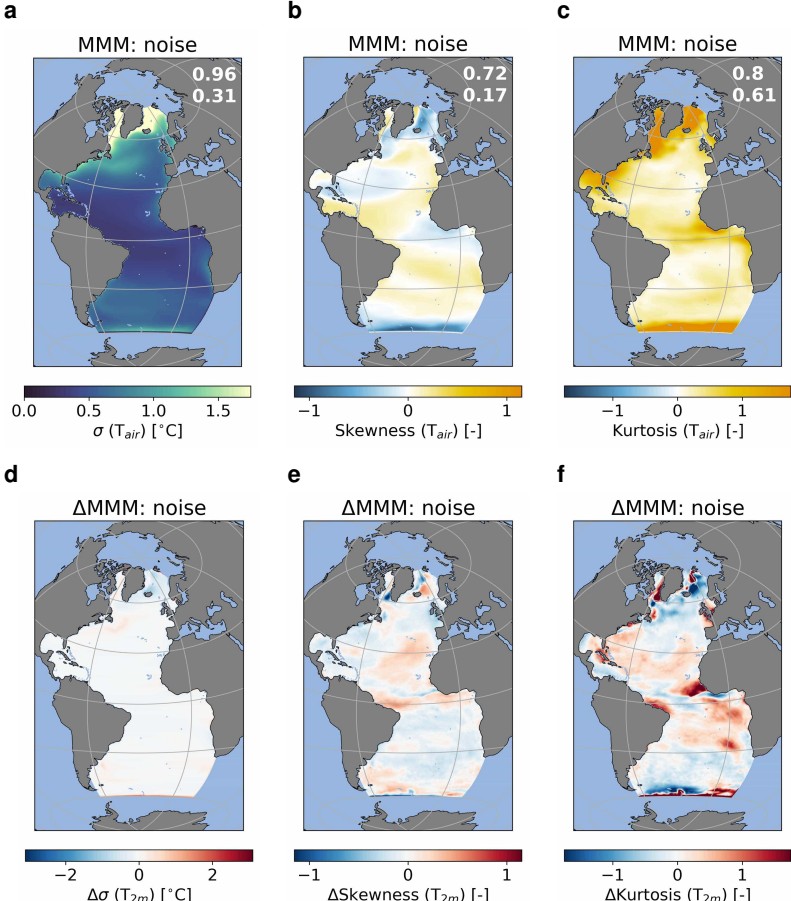

**Figure 4.** As Fig. 3 but for $T_{2m}$ in °C.

3f). In the region between 10°S and 25°N there is a patchy response where most regions see an underestimation of the excess kurtosis (red colors) and some regions an overestimation (blue colors) compared to the ERA5 noise.

The CMIP6 MMM does capture the spatial pattern and amplitude of the standard deviation of the noise in $T_{2m}$ well compared to the ERA5 noise (Fig. 4a,b). The spatial pattern of the skewness is captured reasonably well in the Northern Hemisphere, but the amplitude is typically smaller than in the ERA5 noise (Fig. 4c, d). In the Southern Hemisphere the CMIP6 MMM shows mostly slightly positive skewness, whereas the ERA5 noise mostly shows small negative skewness. The absolute differences are not that large, but there is an important difference in sign. For excess kurtosis the spatial pattern is also relatively similar in the CMIP6 MMM compared to the ERA5 noise, however, the regions in the ERA5 noise with small negative kurtosis are not captured by the CMIP6 MMM (Fig. 4e,f). The amplitude of the excess kurtosis, however, is not as well resolved as the spatial pattern. Most regions in the CMIP6 MMM show an underestimation of the excess kurtosis compared to ERA5.

## 4 Noise model

The CMIP6 MMM appears to do a decent job in capturing the observation-based noise field of both $E - P$ and $T_{2m}$. However, there is still a large spread in the model ensemble, meaning not all models are able to capture these noise fields adequately. Our aim in this section is to develop a statistical model of the noise in both $E - P$ and $T_{2m}$ that can be used as forcing in ocean models. We have tried several methods to construct such a model and we will present four of those below. All these models are based on the ERA5 reanalysis data.

Three of the methods are based on a principal component analysis (PCA) in which we base the noise model on the Principal Components (PCs) and corresponding Empirical Orthogonal Functions (EOFs). The PCA is performed on the noise and is weighted to account for the grid cell areas. For all three methods we use the number of EOFs necessary to explain 90% of the variance in the noise (i.e. 289 EOFs and PCs for $E - P$, and 53 for $T_{2m}$). For the first two methods we directly sample (with replacement) from the PCs. The first method we name PC (1), as we select one random time step (i.e. month) for all PCs. For the PC(1) method we uniformly sample one integer from 1 to the length of the PCs, i.e. 996. We apply this integer for all PCs. For example, if our integer is 7, then we sample the 7th month of each PC to construct the noise model. Using this method we therefore have in total 996 different realizations to sample from, meaning this method is not strictly stochastic. The second method (PC (N)), we sample a random time step out of the PCs, but a different time step for each PC. For the third method (PC (NIG)), we fit a Normal Inverse Gaussian (NIG) distribution to the individual PCs, and next sample randomly from these distributions in a similar fashion as the PC (N) method. The NIG distribution used in the PC (NIG) model has a probability density function determined by

$$f(x, \alpha, \beta, \delta, \mu) = \frac{\alpha \delta K_1(\alpha \sqrt{\delta^2 + (x-\mu)^2})}{\pi \sqrt{\delta^2 + (x-\mu)^2}} e^{\delta \sqrt{\alpha^2 - \beta^2} + \beta(x-\mu)}, \tag{1}$$

Here $\alpha$ is a tail heaviness parameter, $\beta$ an asymmetry parameter, $\mu$ regulates the shift of the distribution, and $\delta$ the scale of the distribution. $K_1$ represents a modified Bessel function of the second kind.

We choose to use three different PCA-based models. The PC (1) model is used to test whether the PCAs can in fact capture the statistics of the noise well. However, since this method is not fully stochastic we also chose to use other models. The PC (N) model is in set-up very similarly, but it has a larger number of values to sample from than the PC(1) method. Since the PC (N) model also has a discrete number of values to sample from, we also used the PC (NIG) model, which does not have this problem. For all three methods, noise fields are constructed by multiplying the value sampled from the PCs with the spatial patterns captured by the EOFs and next summing over the number of PCs/EOFs. Results for the PC (1) and PC (N) models can be found in the Appendix (Fig. A6 to Fig. A9). The PC (NIG) model shows a good agreement with the noise diagnosed from the ERA5 data for the spatial patterns of the standard deviation (Fig. 5a, d), but it is unable to capture the spatial patterns of the skewness (Fig. 5b, e) and excess kurtosis (Fig. 5c, f). The standard deviation in the noise is captured reasonably well (Fig. 5d). Looking at the skewness (Fig. 5e), and the excess kurtosis (Fig. 5f), we can see that this model is unable to represent these metrics correctly, since the PC (NIG) model simulates near zero skewness and excess kurtosis. Just as for the $E - P$ flux, the PC (NIG) model represents the spatial pattern of the noise in the $T_{2m}$ well in the standard deviation (Fig. 6a, d), but not in

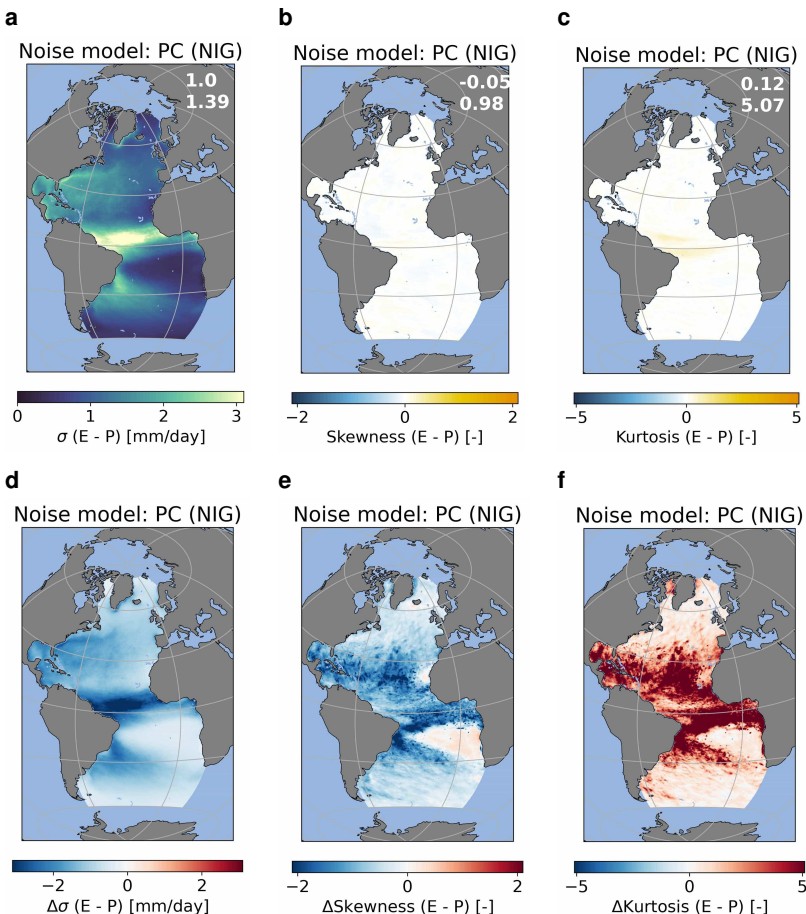

**Figure 5.** Standard deviation ($\sigma$), skewness and excess kurtosis of the noise from the PC (NIG) model for the $E-P$ flux (a) – (c). Differences with ERA5 data (i.e. ERA5 minus PC (NIG)) are shown in (d) – (f). The statistics of the noise model are based on 10000 realizations (months). The numbers in the top right corner of (a) – (c) reflect the spatial correlation and root mean square error. Units for (a) and (d) are mm/day.

the skewness (Fig. 6b, e), and excess kurtosis (Fig. 6c, f). Again, the skewness and excess kurtosis are near zero in all regions, except for the excess kurtosis in the sea ice covered regions in the North Atlantic.

Since the models using the PCA show difficulty in representing the ERA5 noise we have, as the fourth method, also fitted several statistical distributions directly to the noise for each grid cell. For both the $E-P$ and $T_{2m}$, the Normal Inverse Gaussian (NIG) distribution appeared to be the best fit. Note that we have tried several other distributions as well, all of which performed worse than the NIG distribution. Other tested distributions mostly fail to capture the excess kurtosis well. A summary of the performance of a selection of the tested models can be seen in Fig. A18.

We have tested the goodness-of-fit with several measures (Fig. A16). Firstly, we have performed an Anderson-Darling test on normality. We find that for the $E-P$ noise, only $8\%$ of the grid points pass this test (p < 0.05) (Fig. A16b). For $T_{2m}$ this

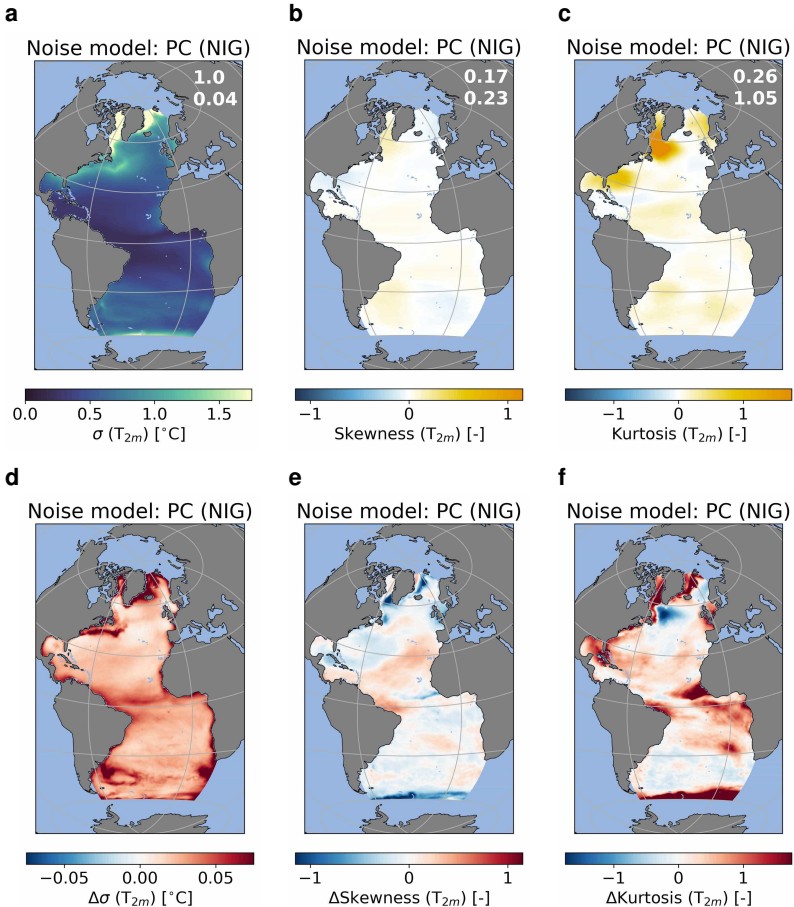

**Figure 6.** As Fig. 5 but for $T_{2m}$ in $°C$

is higher, i.e. $42\%$ (Fig. A16g). Next we have tested whether the NIG provides a better fit than a normal distribution for each grid point. For this we use the following measure:

$$\chi_n = \frac{1}{N} \sum_{i=1}^{N} \frac{(f_i - m_i)^2}{m_i^2}, \tag{2}$$

where N is the number of bins used (i.e. 50), $f_i$ the probability density function of the timeseries per grid point, and $m_i$ the fitted probability density function which is either fitted to an NIG distribution or a normal distribution. We compute $\chi_n$ for both an NIG and Gaussian fit and compare the two. For $98\%$ of the grid points the NIG fit performs better (i.e. $\chi_n$ is smaller for the NIG fit) for the $E - P$ noise (Fig. A16a), and $94\%$ for the $T_{2m}$ noise (Fig. A16f). To test whether the NIG model is a good fit, we apply a Kolmogorov - Smirnov test. For the $E - P$, only 27 grid points do not pass this test, and for $T_{2m}$, 8 grid

points do not (out of 138,788 ocean grid points) (p < 0.05). However, the Kolmogorov - Smirnov test is not well suited for

heavy tailed distributions as we find in our data. Ideally, we would like to perform an Anderson-Darling test, or similar, as a goodness-of-fit test to check whether the NIG fits are statistically significant, but this is computationally too expensive. For the Anderson - Darling test we need to compute critical values which is computationally demanding. Since these critical values are dependent on the parameters of the NIG distribution in Eq. 1 (i.e. $\alpha$, $\beta$, $\mu$ and $\delta$), we would have to repeat the computations for each grid point which leads to high computational cost. As an alternative, we computed the AIC and BIC (Fig. A16c, h) scores for both the Gaussian and the NIG fits. For the $E - P$ noise, the NIG fit is a better fit compared to a Gaussian fit for 98% and 87% of the grid points for the AIC and BIC metrics, respectively. For the $T_{2m}$ noise this is 62% and 35%, respectively. Lastly, we test the significance of the skewness and kurtosis of the $E - P$ and $T_{2m}$ noise. We do this by fitting an AR(1) model to the data, and subsequently generate sampling statistics from this model. The fitted (Gaussian) AR(1) model fails to represent the skewness and excess kurtosis in the $E - P$ noise (p < 0.05) for 93% and 85% of the grid points (Fig. A16d, e), and 38% and 53% of the grid points for the $T_{2m}$ skewness and excess kurtosis (Fig. A16i, j). .

Based on this collection of tests, we think that for most of the grid points the NIG model provides a good fit to the data. Furthermore, following the Anderson-Darling test on normality and the fitted AR(1) model, most of the $E - P$ noise is non-Gaussian, and to a lesser degree this also applies to the $T_{2m}$ noise. The grid points for the $E - P$ noise that are likely Gaussian are located in clusters 4 and 11, which are indeed clusters with skewness and excess kurtosis close to 0 (Fig. 2a, b). For the $T_{2m}$ noise the grid points that show Gaussian behavior are mainly located in the sea-ice free subpolar Ocean. These grid points mainly belong to clusters 1, 7 and 12 which are also the clusters with approximately zero skewness and near-zero excess kurtosis (Fig. 2c, d).

Using the fitted NIG distribution, we can generate a fully stochastic noise field for each month using the 4 parameters per grid cell. The model shows a good agreement with the noise diagnosed from the ERA5 data for the spatial patterns of the standard deviation (Fig. 7a, d), skewness (Fig. 7b, e) and excess kurtosis (Fig. 7c, f). Especially the standard deviation in the noise is captured well with only small deviations between 10°S and 25°N (Fig. 7d). The NIG distribution underestimates the regions with strong negative skewness over the latitude bands 0°N to 10°S and 10°N to 25°N (Fig. 7e). For excess kurtosis we see a similar underestimation in these regions, meaning that the excess kurtosis is higher in the ERA5 data (Fig. 7f). However, the region between these two latitude bands shows a much higher excess kurtosis in the NIG model compared to the ERA5 noise.

Just as for the $E - P$ flux, the NIG model represents the spatial pattern of the noise in the $T_{2m}$ well in the standard deviation (Fig. 8a, d), skewness (Fig. 8b, e), and excess kurtosis (Fig. 8c, f). Also here the standard deviation is captured very well by the NIG model with only very small differences in the sea ice covered regions (Fig. 8d). The same applies to the skewness, where we also see some deviations in these same regions (Fig. 8e). For most regions the NIG model captures the excess kurtosis quite well (Fig. 8f). However, for regions with a high excess kurtosis in the ERA5 noise, such as the sea ice covered regions and the Gulf of Mexico, the NIG model strongly overestimates the excess kurtosis.

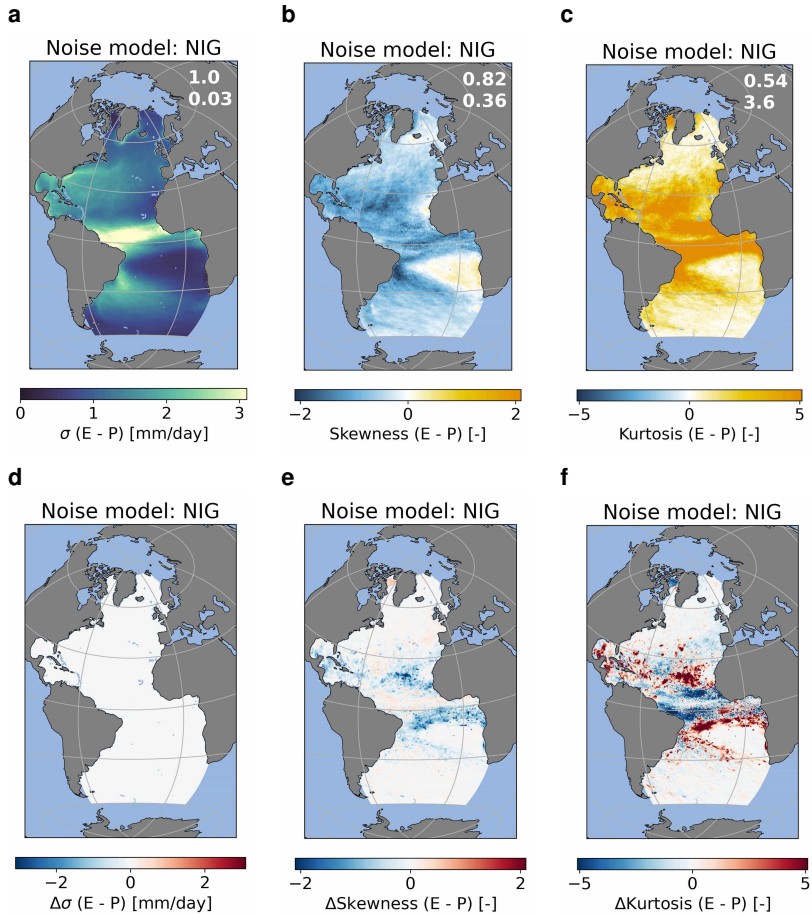

**Figure 7.** Standard deviation ($\sigma$), skewness and excess kurtosis of the noise from the NIG model for the $E - P$ flux (a) – (c). Differences with ERA5 data (i.e. ERA5 minus NIG) are shown in (d) – (f). The statistics of the noise model are based on 5000 realizations (months). The numbers in the top right corner of (a) – (c) reflect the spatial correlation and root mean square error. Units for (a) and (d) are mm/day.

## 5 Performance CMIP6 and NIG models

In this section, we compare the noise models and the CMIP6 models with the ERA5 noise using Taylor diagrams (Fig. 9) to
provide a more in-depth discussion on the performance of the individual models. We compare how well the different models represent the standard deviation (Fig. 9a, b), the skewness (Fig. 9c, d), and the excess kurtosis (Fig. 9e, f) found in the ERA5 noise. Taylor diagrams are a good tool to better understand the performance of all the different models against the observation-based noise. In a Taylor diagram, three metrics are displayed: (1) the spatial correlation coefficient, (2) the variation in the data as represented by the standard deviation, and (3) the root mean square error between the observation-based data and the
model. The spatial correlation coefficient is displayed on the outer circle in the figure and the straight dotted lines in Fig. 9,

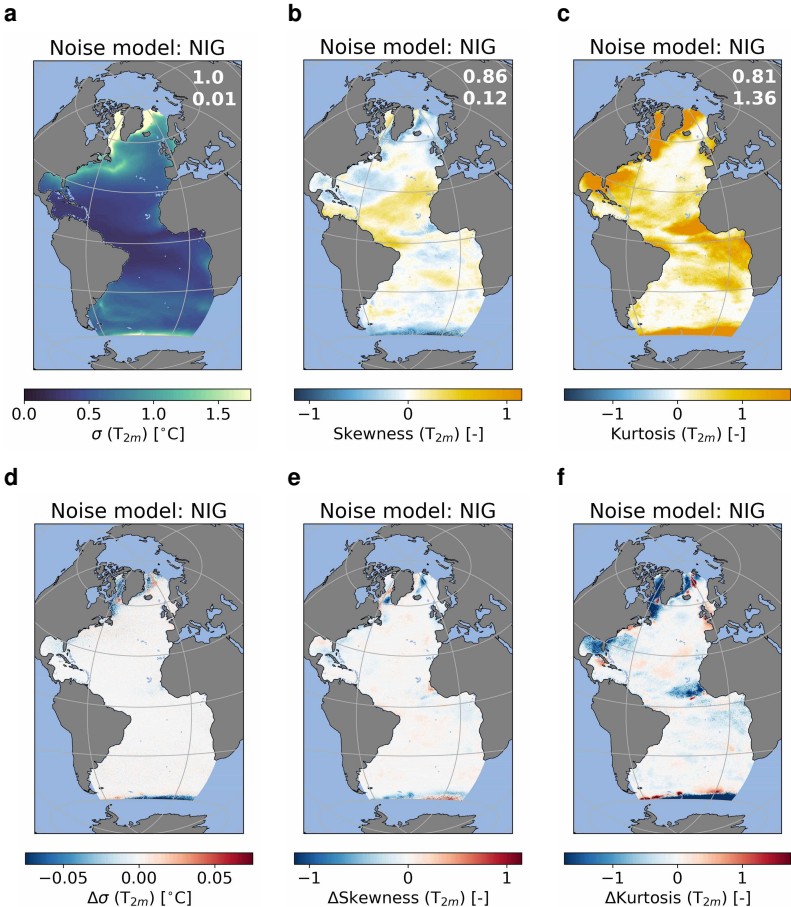

**Figure 8.** As Fig. 7 but for $T_{2m}$ in °C.

connecting the origin with the outer circle are lines of constant correlation. The standard deviation is displayed on both the x-axis and y-axis. Lines of equal standard deviation are circles with their center in the origin of the plot. The black dashed line in Fig. 9 displays the standard deviation in the observation-based noise. The RMSE is displayed with the black contour circles with their center in the observation-based noise marker. The location of each of the markers therefore provides information about three important metrics and therefore the performance of the individual models compared to the observation-based noise. Ideally, a model will be in the lower part of the graph, since this indicates high spatial correlation, close to the black dashed lines, since this indicates similar variability compared to the observation-based noise, and by combining these two the RMSE will consequently also be low. All three metrics are determined using weights considering the area of each grid cell.

For the $E - P$ noise models, the PC (1) model performs best for the standard deviation, skewness and excess kurtosis and the PC (NIG) the worst (Fig. 9 a, c and e). The PC (N) model performs equally well for the spatial correlation, but strongly underestimates the variability in skewness and excess kurtosis. The NIG model has a lower spatial correlation, but is much

better in capturing the variability in all three statistical moments. All models have trouble representing the excess kurtosis in the latitudinal bands 10°S to 25°N. The PC (1) and PC (N) models overestimate the excess kurtosis in almost the entire region, whereas the NIG model underestimates the excess kurtosis over the ITCZ region and overestimates it in the other regions. This is because this region can experience very extreme rainfall episodes with a very low number of occurrences which severely affects the excess kurtosis diagnosed from the ERA5 noise as was also found with the clustering analysis (Fig. 2). Because these episodes only occur a few times in the time series, these are not represented well by the NIG model, and are also difficult to represent in the PC (1) model.

We can explain the failure of the PC (NIG) model to accurately resemble the observation-based skewness and excess kurtosis by the Central Limit Theorem. This theorem states that when summing over random variables, the distribution of this sum converges towards a Gaussian distribution, which, by definition, has zero skewness and excess kurtosis. What we do in these PC-models is that we sample values from the PCs, multiply those with the EOFs and sum these, which, following Central Limit Theorem, converges towards a Gaussian distribution. The same applies to the PC (N) model which performs well for spatial correlation skill, but (based on a timeseries of 10,000 realizations) underestimates the amplitude of the skewness and excess kurtosis. This underestimation increases when longer timeseries are used, and the model slowly converges to a Gaussian one. Methods based on a PCA, except for the PC (1) model, will therefore be unable to represent the skewness and excess kurtosis in the observation-based noise. An alternative explanation as to why the PC-based models fail to capture the skewness and excess kurtosis is that the PCs might be (non-linearly) dependent on each other. To test this, we have calculated the distance correlation (Székely et al., 2007) between the PCs, including whether the distance correlation is significant (p-value < 0.05) based on a permutation test of n = 1000 (Fig. A17). For both the $E - P$ and $T_{2m}$ PCs, around 5% of the possible PC combinations experiences a significant dependence. However, the strongest distance correlation is only 0.14 for the PCs corresponding to the $E - P$ noise, and 0.11 for the PCs corresponding to the $T_{2m}$ noise, meaning there is at best a very weak dependence between the PCs. We therefore do not expect that the weak non-linear dependence between some of the PCs is the reason why the PC-based models fail, but that the explanation mentioned before, i.e. the Central Limit Theorem, is the main reason.

For $T_{2m}$, Fig. 9 b, d and f show that the NIG and PC (1) models perform consistently best. All models capture the spatial pattern in the standard deviation of the noise as shown by the near unity spatial correlation coefficient, however, the PC (N) and PC (NIG) models both overestimate the variability in the standard deviation of the noise as shown by the high RMSE and larger standard deviation (Fig. 9b). The spatial pattern of the skewness is captured reasonably well by the NIG, PC (1) and PC (N) models, but not by the PC (NIG) model (Fig. 9d). The PC (N) shows a stronger underestimation of the variability compared to the PC (1) and NIG model. For the excess kurtosis a similar conclusion can be drawn, except that the NIG model strongly overestimates the variability (Fig. 9f). The worse performance for excess kurtosis can be explained by the overestimation of the sea ice covered regions and the Gulf of Mexico by the NIG model. In these regions, the distribution of the ERA5 noise has a relatively broad, flat peak or sometimes a slightly bimodal peak. This is the reason the NIG fit does not perform very well in these regions. Similar as to the $E\breve{\ }P$ noise, the PC (N) and PC (NIG) models are unable to explain the variability in the skewness and excess kurtosis as explained above.

As discussed in Section 3.2, the CMIP6 MMM captures the ERA5 $E - P$ noise reasonably well, though the performance decreases for the higher statistical moments. This is likely related to strong biases over the South Equatorial Current where the skewness is too negative in the CMIP6 MMM, and the excess kurtosis too positive compared to the ERA5 noise. This is potentially related to the double ITCZ bias present in most CMIP6 models (Tian and Dong, 2020). The latitudinal extent of the ITCZ is too southward in many models, which also causes a shift in the higher order statistical moments in this region resulting in relatively large biases. From Fig. 9 we see that the individual models that consistently perform the best are CESM2-WACCM (30), CESM2 (31) and NorESM2-MM (32) (except for excess kurtosis where NorESM2-MM has quite a large RMSE). What these models have in common is that their atmospheric model is the Community Atmosphere Model 6 (CAM6), or in the case of CESM2-WACCM based on CAM6 and run on a nominal 1° horizontal resolution. This suggests that this atmospheric model is able to capture the observation-based noise reasonably well.

Liu et al. (2022) also found that these models are performing relatively well for precipitation biases which they suggest is due to the specific two-moment prognostic cloud microphysics scheme (Gettelman and Morrison, 2015) used in CAM6. TaiESM1, which uses CAM5 and an earlier version of the prognostic cloud microphysics scheme also performs relatively well. There are also two other CESM2 models that use a form of CAM6, i.e. CESM2 – WACCM – FV2 (27) and CESM2 – FV2 (29). These models perform less well as the other three, which might be explained by the fact that these models are run on a lower (i.e. 2°) resolution. The CMIP6 MMM has the same biases in the latitudinal band between 10°S and 25°N, though less strong in some regions. This is probably because the high rainfall episodes in the ERA5 data are smoothed when regridded to a 1° grid, which is done before comparing it to the CMIP6 models and MMM.

For $T_{2m}$ the CMIP6 MMM also performs reasonably well compared to the ERA5 noise, and just as for the $E - P$ noise, performance is lower for higher statistical moments. The strongest biases (both positive and negative) for the excess kurtosis are found over the sea ice covered regions. This might be related to biases in sea-ice cover in the CMIP6 models (Watts et al., 2021). For the individual models it is more difficult to point towards consistently well performing models. The UKESM1-0-LL (22) model simulations performed by the MOHC shows the most consistency. Other models that perform relatively well in 2 out of 3 statistical moments are CESM2-FV2 (29) and CAS-ESM2-0 (8). Interestingly, the UKESM1-0-LL (33) simulations performed by the NIMS-KMA are among the worst performing models. The only difference between the two models is the computer on which the model is run on, and the initial conditions. This suggests that there is also a dependency on initial conditions in the performance of the CMIP6 models.

Except for the excess kurtosis in the $T_{2m}$ noise, the NIG model outperforms the individual CMIP6 models and MMM which is due to the overestimation of the excess kurtosis over sea ice covered regions by the NIG model. The PC (NIG) model only outperforms the CMIP6 MMM for the standard deviation and is very poor for the skewness and excess kurtosis. The PC (1) model outperforms the CMIP6 models and MMM for the skewness and excess kurtosis. The PC (N) model outperforms the CMIP6 MMM for all moments with respect to the spatial correlation and is very similar to the CMIP6 MMM and the best CMIP6 models for RMSE. This means that we can capture 'realistic' noise better with the statistical model than the fully coupled Earth System Models. Among the PC-based models, the PC (1) model performs best and similar to the NIG model, but this model is not fully stochastic as the other noise models.

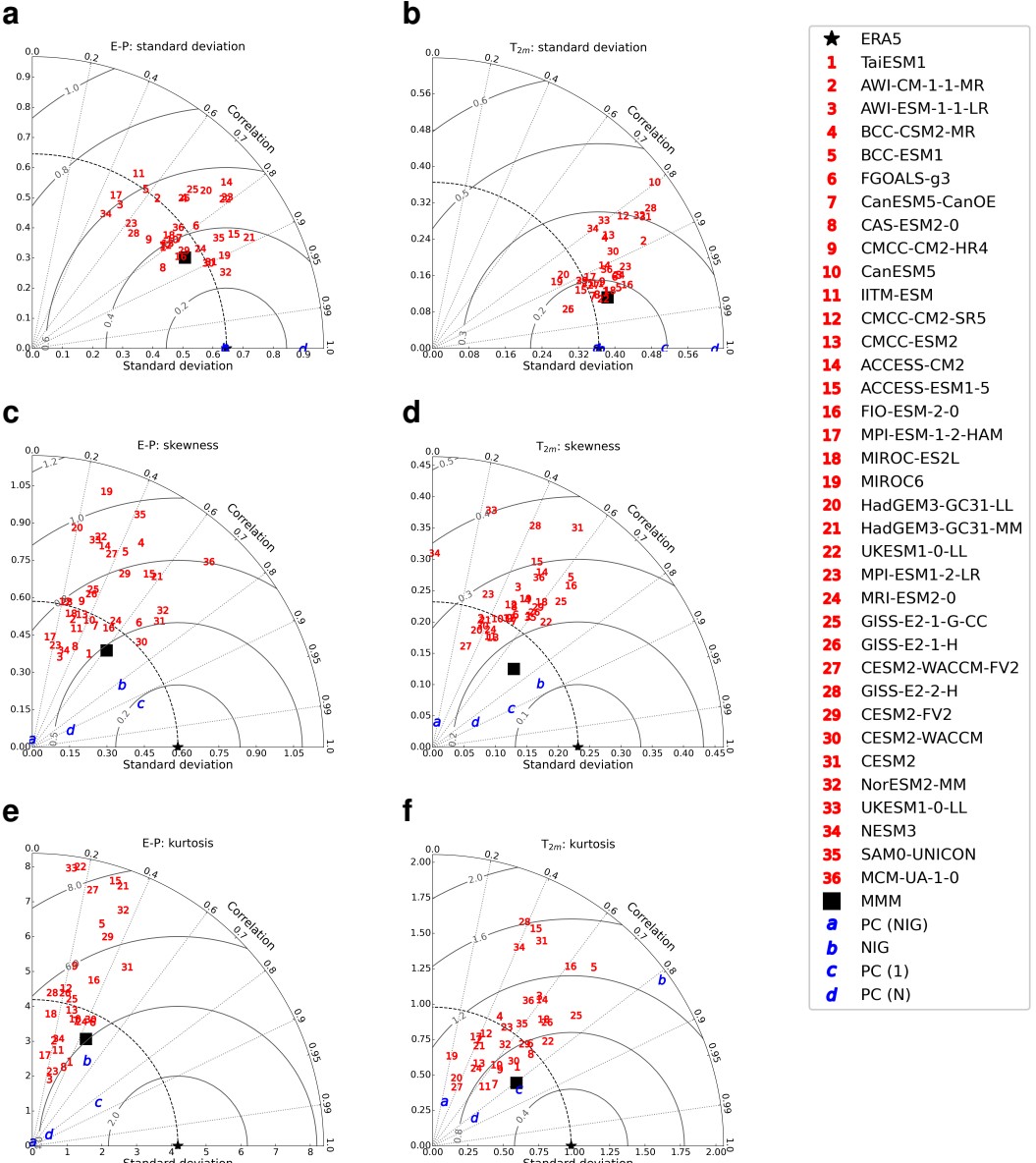

**Figure 9.** Taylor diagrams for statistics of the noise. (a) and (b) standard deviation. (c) and (d) skewness, and (e) and (f) excess kurtosis. (a), (c) and (e) are for the E – P noise, and (b), (d) and (f) for the noise in the $T_{2m}$. The star refers to the ERA5 data, the different red numbers refer to the different CMIP6 models, the blue letters to the noise models, and the square black marker represents the CMIP6 MMM. Note that UKESM1-0-LL number 22 is performed by the MOHC and number 33 by NIMS-KMA. Units for standard deviation in (a) are mm/day, and °C in (b).

## 6  Summary and discussion

In this study we have analysed ERA5 evaporation minus precipitation ($E-P$) and 2 m air temperature ($T_{2m}$) fields to determine what observation-based noise is in these variables. We find that due to nonzero skewness and excess kurtosis, the noise in both variables typically cannot be classified as white and studies that assume white noise in either of the two variables might not resolve the response of the ocean to atmospheric noise realistically. We have analysed the noise in 36 different CMIP6 Earth System Models and the CMIP6 multi-model mean (MMM) and compared those to the ERA5 noise. There is quite a spread in the performance of the CMIP6 models, but the MMM is performing relatively well compared to the individual models. Typically, the models perform best for the standard deviation and worst for the excess kurtosis. Furthermore, we have fitted a Normal Inverse Gaussian (NIG) distribution to the ERA5 noise of both variables. This results in a stochastic noise model that can be used as input in Ocean General Circulation Models (OGCMs). We have shown that the NIG model captures the standard deviation, skewness and excess kurtosis of the ERA5 noise reasonably well in both the $E-P$ and $T_{2m}$ except for the excess kurtosis in the $T_{2m}$ noise where the NIG model strongly overestimates the positive excess kurtosis in sea ice covered regions. For most metrics and statistics, the NIG model performs better than the individual CMIP6 models and CMIP6 MMM.

Previous studies have looked into biases in CMIP6 models. However, these studies typically look into the biases in the mean state or the seasonality of the variables. Here, we have specifically looked at variability up to interannual timescales and specifically the distribution and related metrics (i.e. standard deviation, skewness and excess kurtosis). We found that biases in these quantities are still to some extent connected to biases in the mean state. For example, the biases in skewness and excess kurtosis in the $E-P$ noise in the South Atlantic are for example likely to be related to the double ITCZ bias described in earlier studies (Tian and Dong, 2020; Li et al., 2020). Differences in the excess kurtosis in sea ice covered regions can also be related to the biases in Arctic sea-ice thickness and cover (Watts et al., 2021).

In the development of a noise model the best variant turned out to be a point wise statistical fit of a Normal Inverse Gaussian (NIG) distribution. As shown in Section 3, the model performs relatively well in most grid points, but can still deviate quite a bit for especially the excess kurtosis. One major drawback of fitting a statistical distribution point wise to the data is that for the individual noise fields (i.e. one random realization) we lose spatially coherent structures, and potentially auto-correlation in the noise. We have constructed alternative models based on a principal component analysis (PCA) where the corresponding Empirical Orthogonal Functions (EOFs) capture the spatial structures. The PCs contain non-linear effects, but these are difficult to extract statistically. The PC-based models underestimate the skewness and excess kurtosis in the noise fields because of the Central Limit Theorem, or (for the PC (1) model) are not fully stochastic. . Therefore, we eventually decided to fit a model to the data that can relatively accurately represent the standard deviation, skewness and excess kurtosis in the ERA5 noise. However, when the spatially coherent structures captured by the EOFs are deemed more important than an accurate representation of the skewness and kurtosis of the noise, PC-based models can be used. The loss of spatially coherent structures can be important when studying noise-induced transitions of the AMOC. Noise that is spatially coherent influence larger areas of ocean. This could, for example, mean that a freshening of the surface ocean could happen over a larger area of the ocean and therefore

might be more efficient in inhibiting deep convection in the North Atlantic. Whether it is actually important should be tested in an ocean model but this is outside the scope of this study.

Similar studies that look into the characteristics of $E - P$ and $T_{2m}$ noise are sparse. In Sura and Sardeshmukh (2008), they investigate the non-Gaussianity of daily SST variability. The timescales assessed in Sura and Sardeshmukh (2008) are faster (i.e. daily versus monthly), and they look at SSTs, whereas we look at air temperatures. However, relatively similar results are achieved in our study compared to Sura and Sardeshmukh (2008). Skewness in daily SST variability is typically negative in the Atlantic Ocean, whereas the excess kurtosis is mostly positive, similar to what we find for the air temperature. They relate this to multiplicative noise in mixed layer dynamics. However, they make the assumption that daily fluctuations in air temperature are Gaussian. Our study shows, that at least on monthly timescales, this is not the case over most of the ocean. Whether the multiplicative noise signal we find in the $T_{2m}$ noise originates from SST variability or atmospheric dynamics, or a combination of the two, is left for further study.

To conclude, we have provided an analysis of observation-based noise from ERA5 reanalysis data. Based on this realistic noise we have constructed a noise model based on a Normal Inverse Gaussian distribution fit to the ERA5 noise. This product is made publicly available in the repository related to this paper (Boot and Dijkstra, 2024). The noise model can, for example, be used as a forcing on ocean models to study noise-induced transitions of the AMOC under 'realistic' noise forcing.

*Code and data availability.* ERA5 data can be downloaded from the Copernicus Climate Data Store (CDS). CMIP6 data can be downloaded from the Earth System Grid Federation (ESGF) or using the scripts in the repository (Boot and Dijkstra, 2024). Directions on which exact data needs to be downloaded and all scripts used for analyses and making the figures can be found at Boot and Dijkstra (2024). Here also a script that contains the noise models can be found.

*Author contributions.* AAB and HAD conceptualized the study. AAB acquired the results. Both authors contributed to writing the manuscript.

*Competing interests.* The authors declare that they have no conflict of interest.

*Financial support.* This research has been supported by the European Research Council through the ERC-AdG project TAOC (PI: Dijkstra, project 101055096).

.

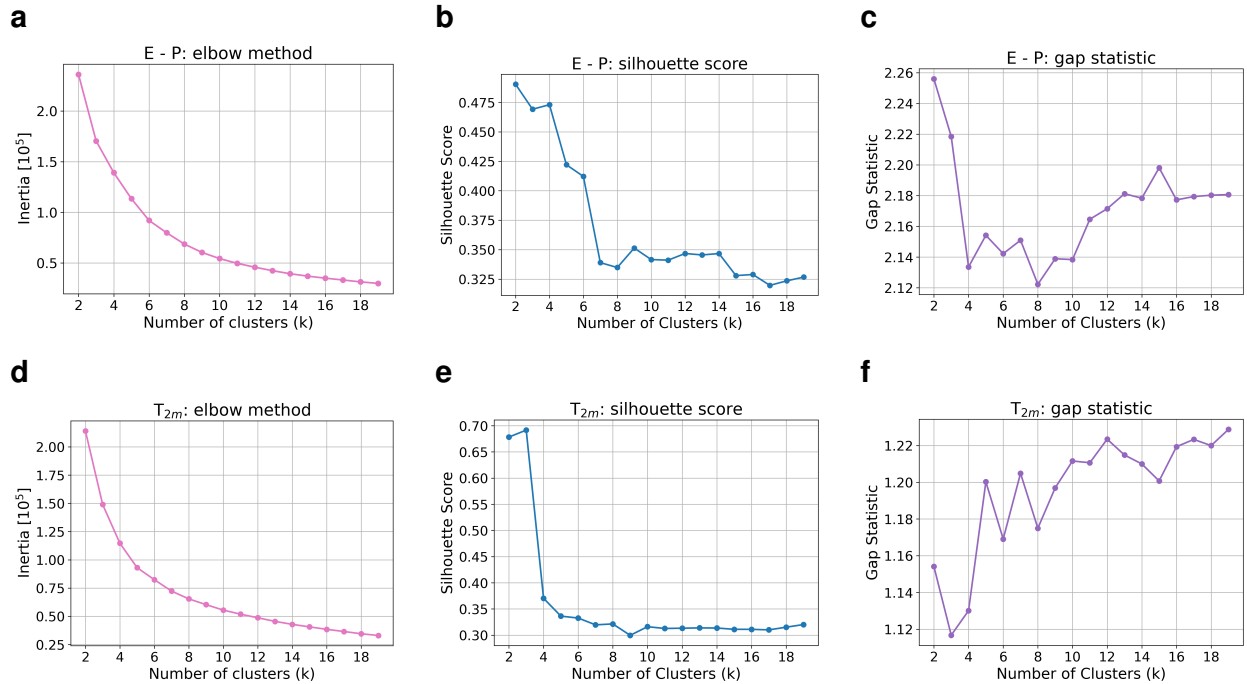

**Figure A1.** Metrics for the k-means clustering method versus the number of clusters for the $E - P$ clusters (a-c) and the $T_{2m}$ clusters. (a) and (d) represent the elbow method, (b) and (e) represent the silhouette score, and (c) and (f) the gap statistic.

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

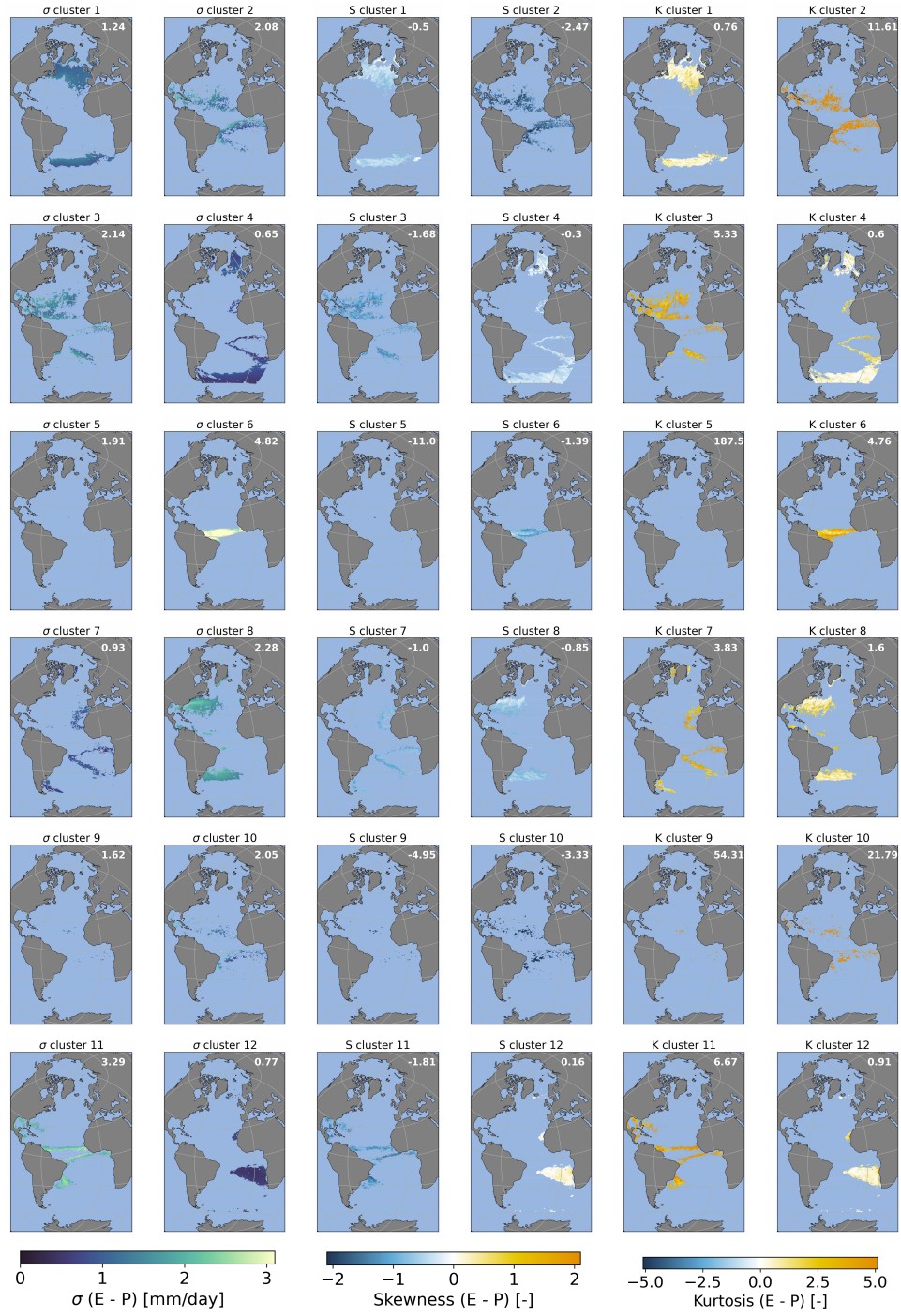

**Figure A2.** The 12 clusters for the $E - P$ noise fields. The columns 1 and 2 correspond to the standard deviation of the clusters, columns 3 and 4 represent the skewness, and columns 5 and 6 excess kurtosis. Numbers in the top right represent area weighted mean of the metric.

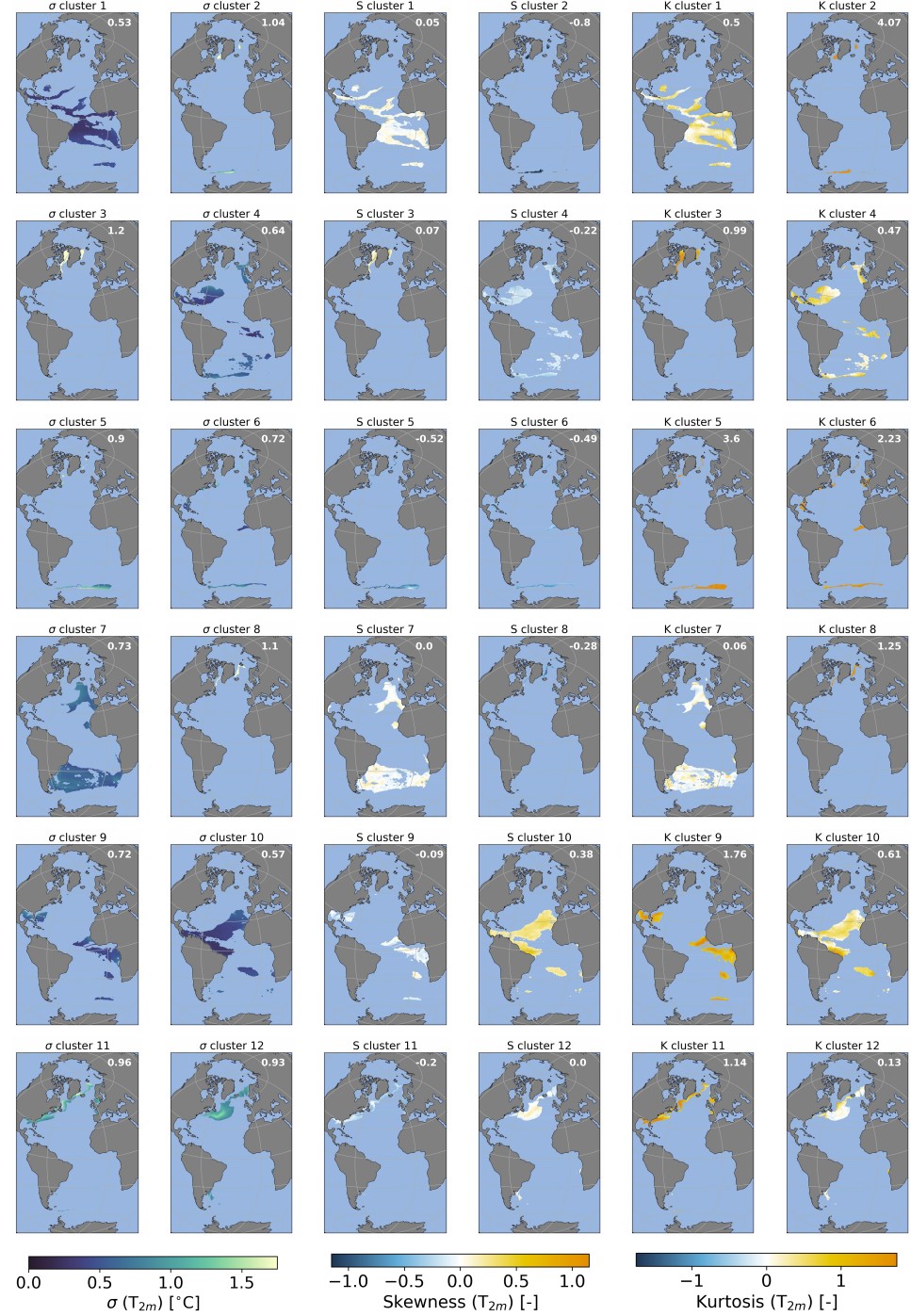

**Figure A3.** As Fig. A2 but for the $T_{2m}$ clusters.

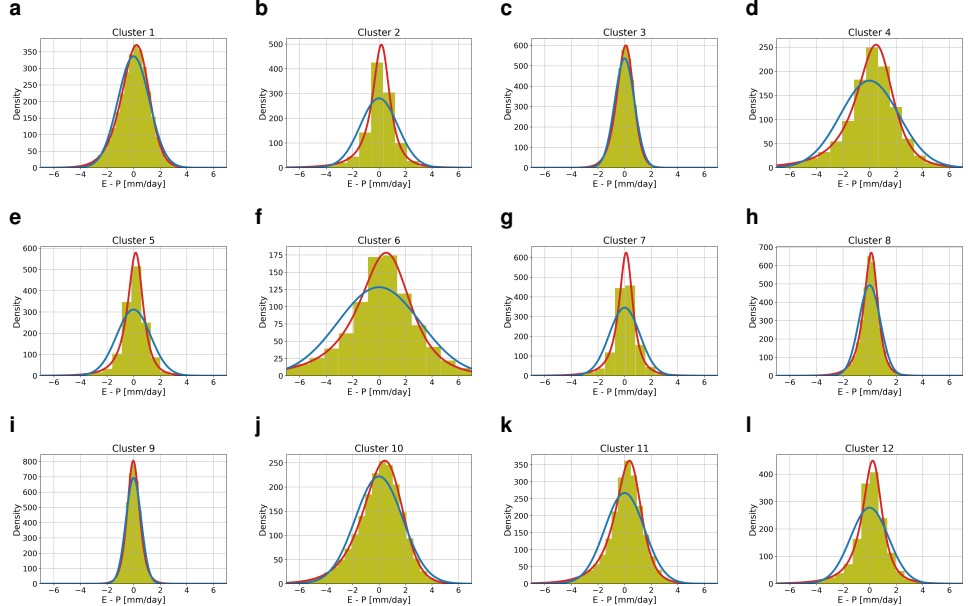

**Figure A4.** Probability density functions for the 12 clusters for the $E - P$ noise. Red lines represent a Normal Inverse Gaussian fit, blue lines a Gaussian fit, and the yellow histrogram the data (using 50 bins). The y-axis shows the density, and the x-axis the $E - P$ noise in mm/day.

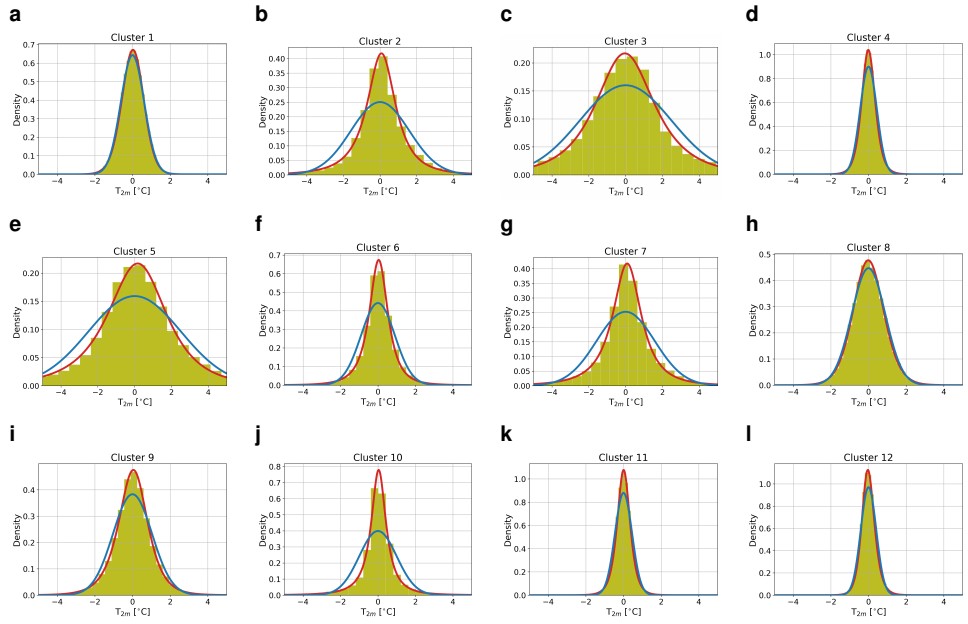

**Figure A5.** As Fig. A4 but for the $T_{2m}$ clusters in °C instead of mm/day.

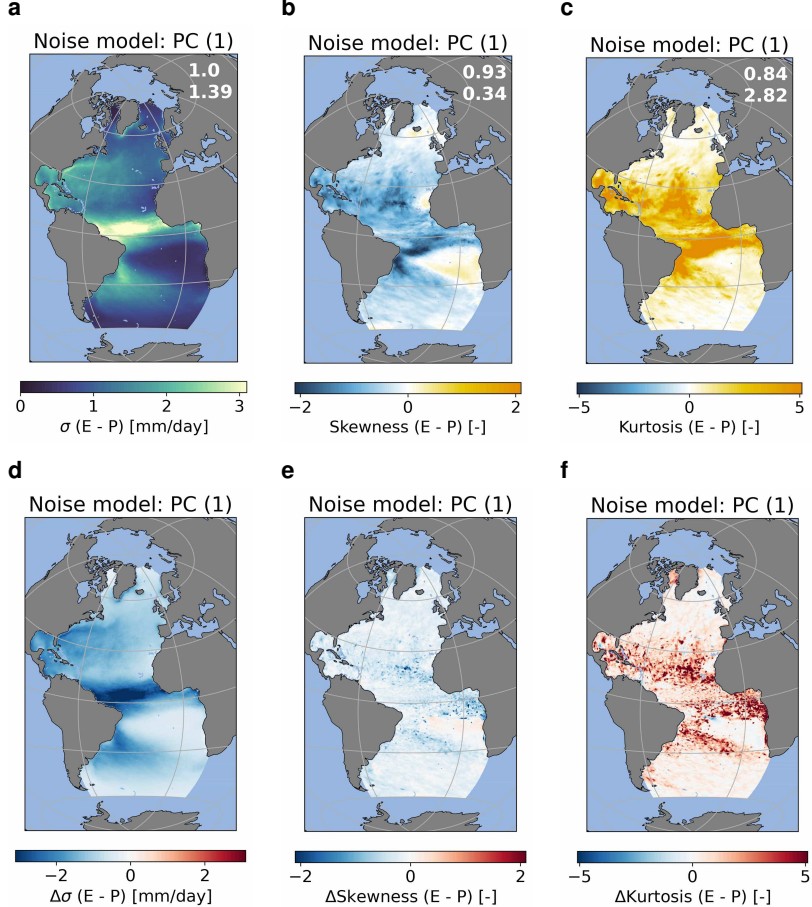

**Figure A6.** Standard deviation ($\sigma$), skewness and excess kurtosis of the noise from the PC (1) model for the $E - P$ flux (a) – (c). Differences with ERA5 data (i.e. ERA5 minus PC (1)) are shown in (d) – (f). The statistics of the noise model are based on 10000 realizations (months). The numbers in the top right corner of (a) – (c) reflect the spatial correlation and root mean square error. Units for (a) and (d) are mm/day.

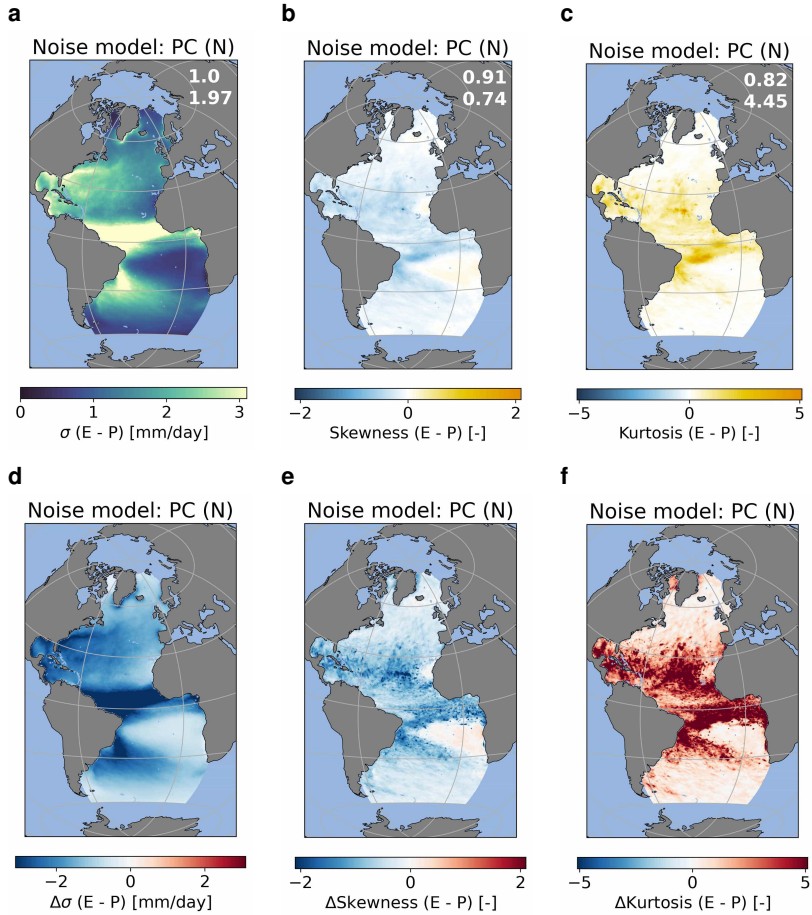

**Figure A7.** As Fig. A6 but for $T_{2m}$.

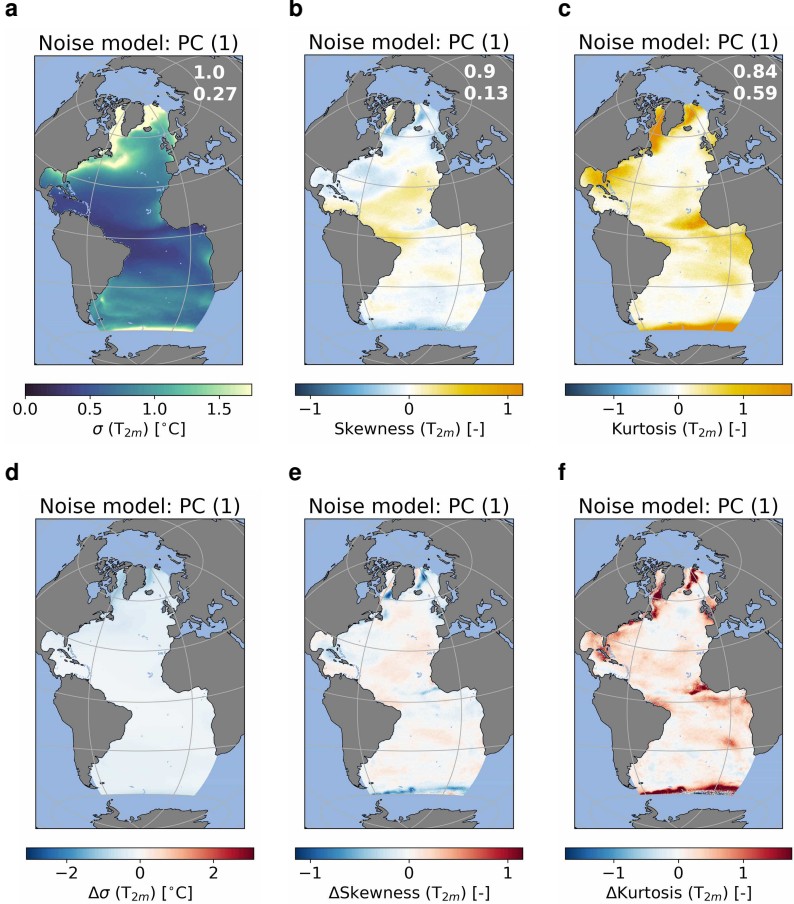

**Figure A8.** Standard deviation ($\sigma$), skewness and excess kurtosis of the noise from the PC (N) model for the $E - P$ flux (a) – (c). Differences with ERA5 data (i.e. ERA5 minus PC (N)) are shown in (d) – (f). The statistics of the noise model are based on 10000 realizations (months). The numbers in the top right corner of (a) – (c) reflect the spatial correlation and root mean square error. Units for (a) and (d) are mm/day.

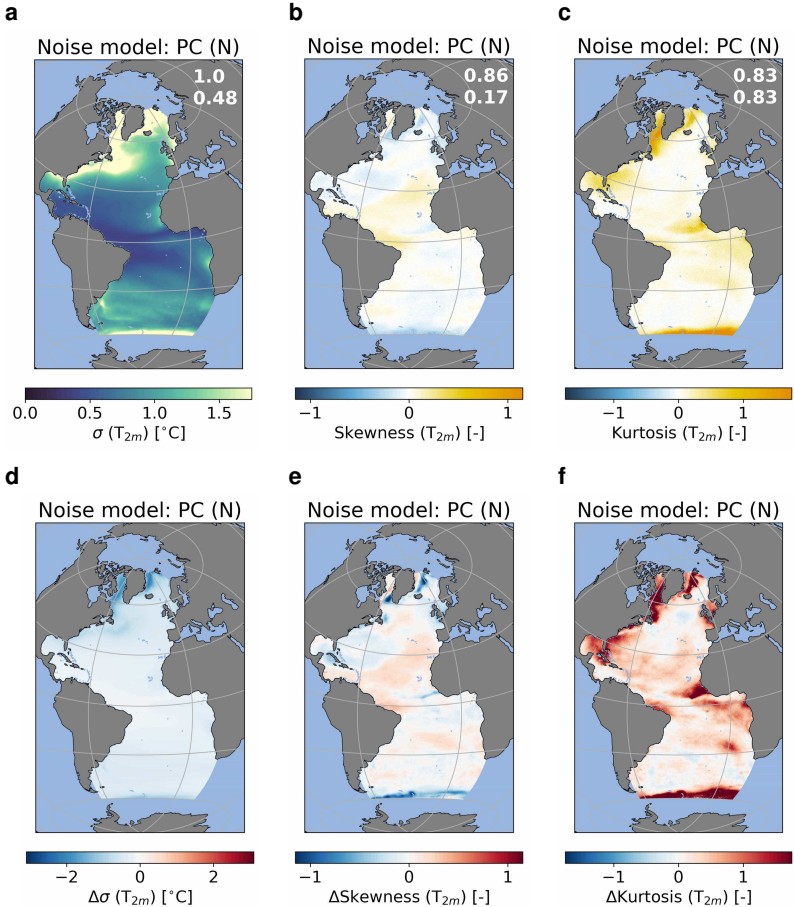

**Figure A9.** As Fig. A8 but for $T_{2m}$.

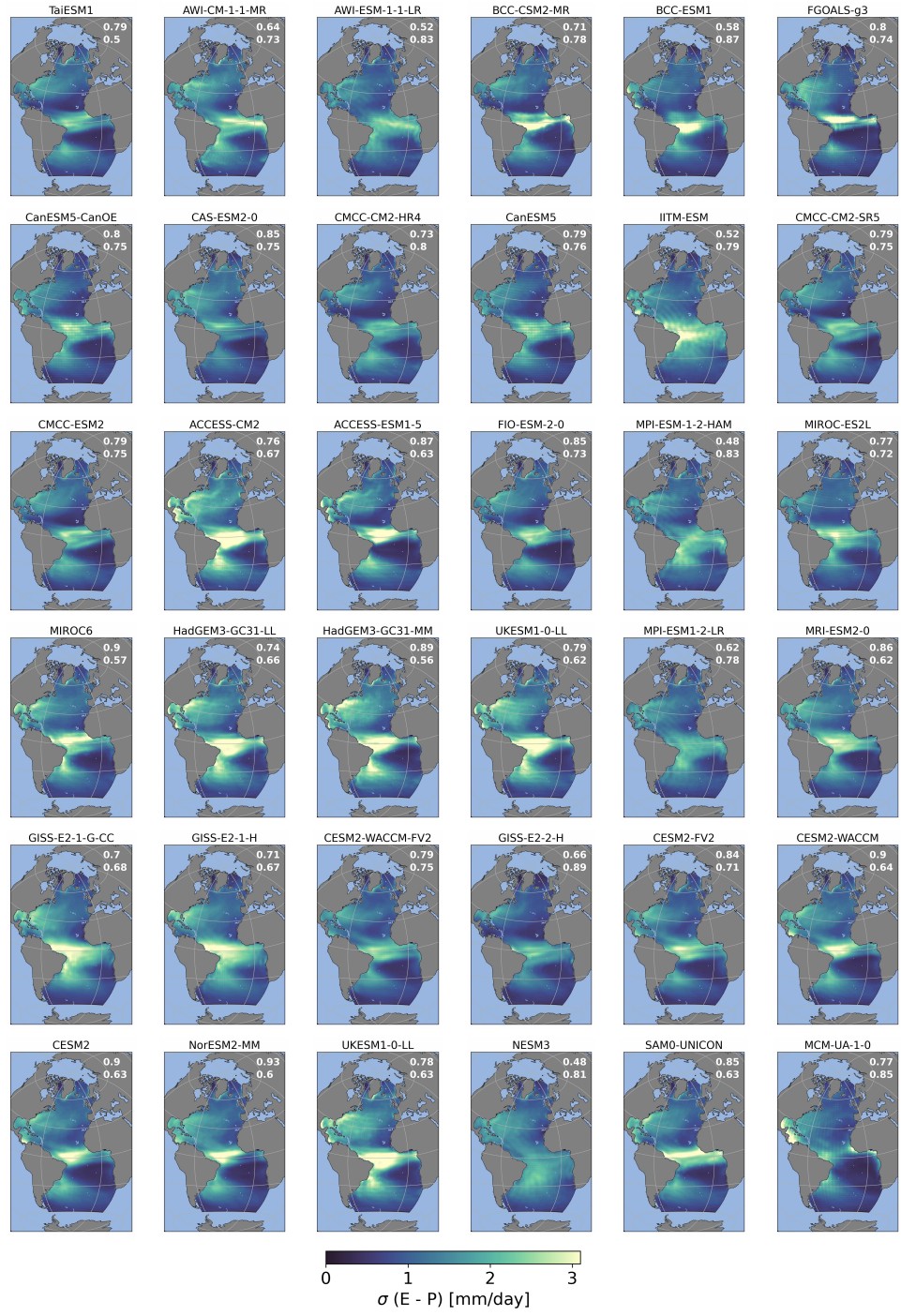

**Figure A10.** Standard deviation ($\sigma$) in the noise of the $E - P$ for the analysed CMIP6 models. Numbers in the top right corner reflect the spatial correlation and root mean square error.

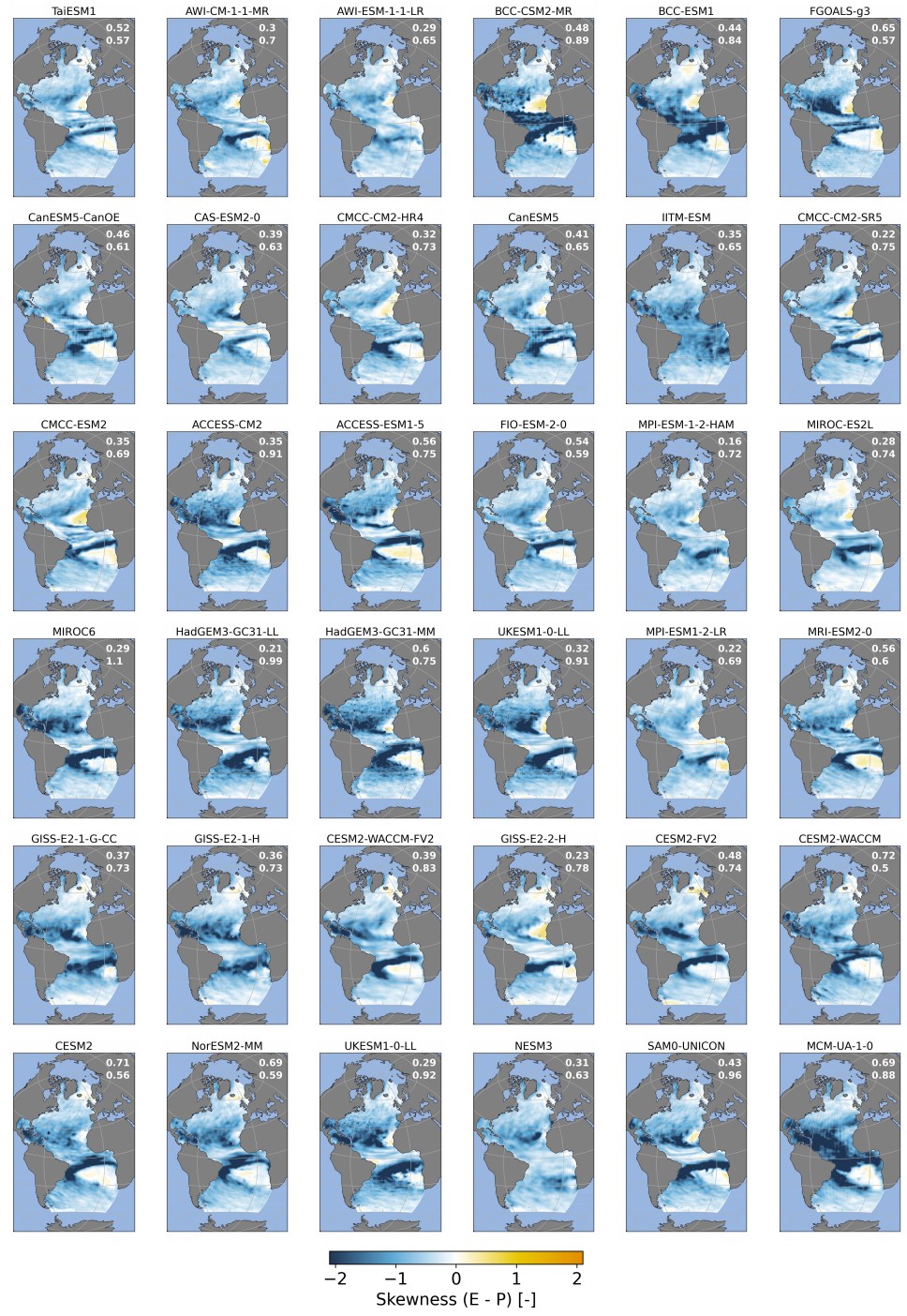

**Figure A11.** Skewness in the noise of the $E - P$ for the analysed CMIP6 models. Numbers in the top right corner reflect the spatial correlation and root mean square error.

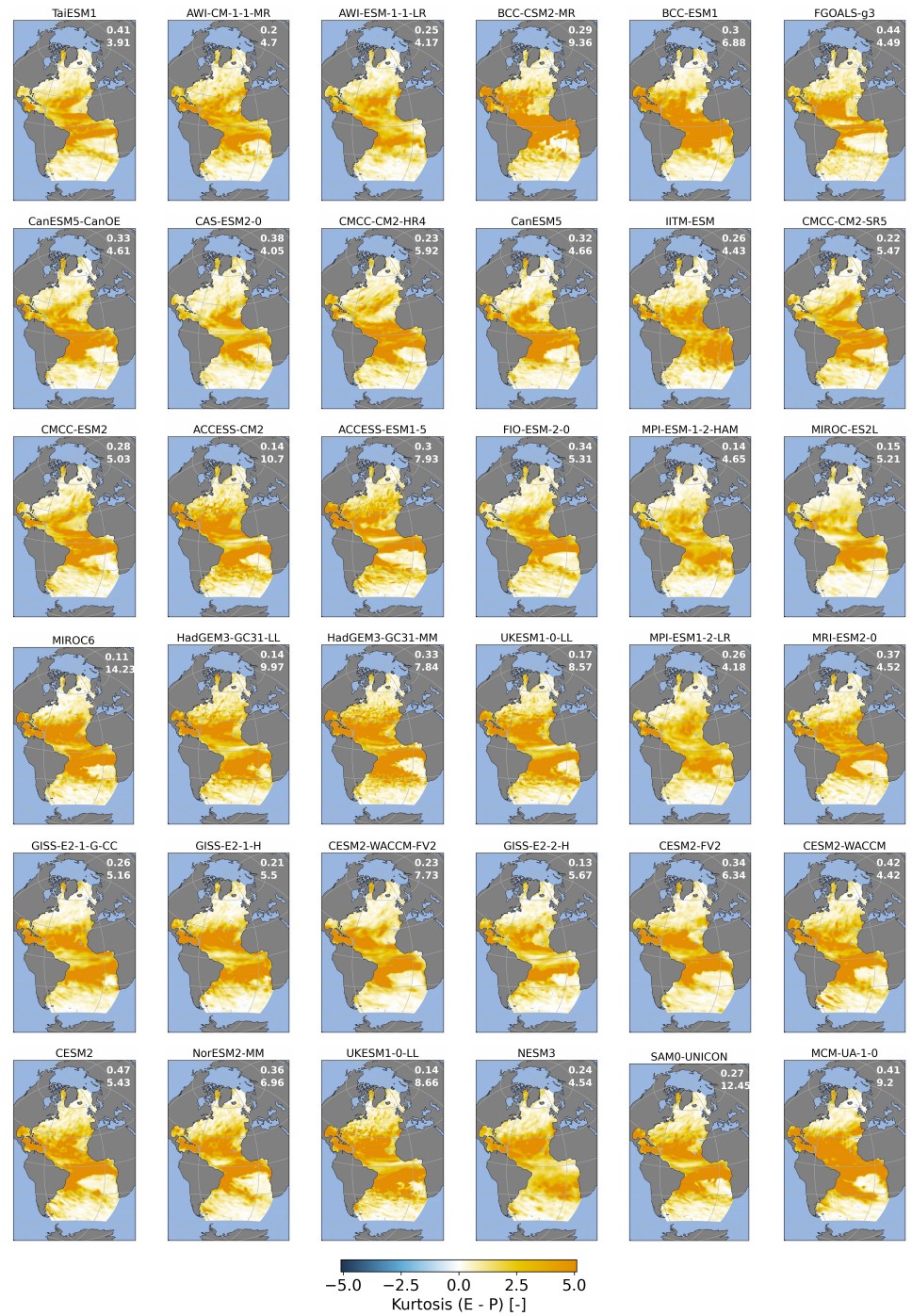

**Figure A12.** Excess kurtosis in the noise of the $E - P$ for the analysed CMIP6 models. Numbers in the top right corner reflect the spatial correlation and root mean square error.

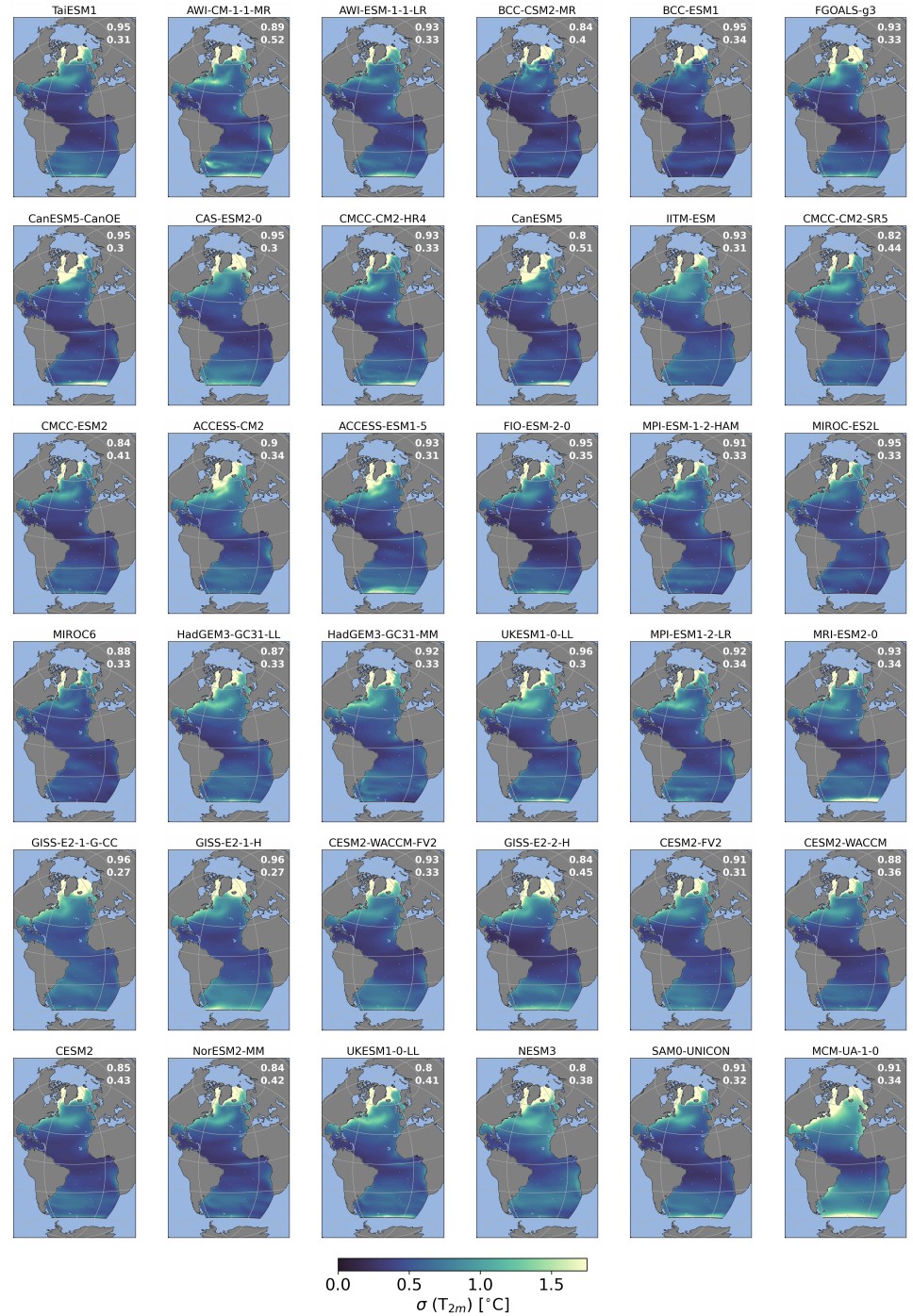

**Figure A13.** As Fig. A10 but for $T_{2m}$.

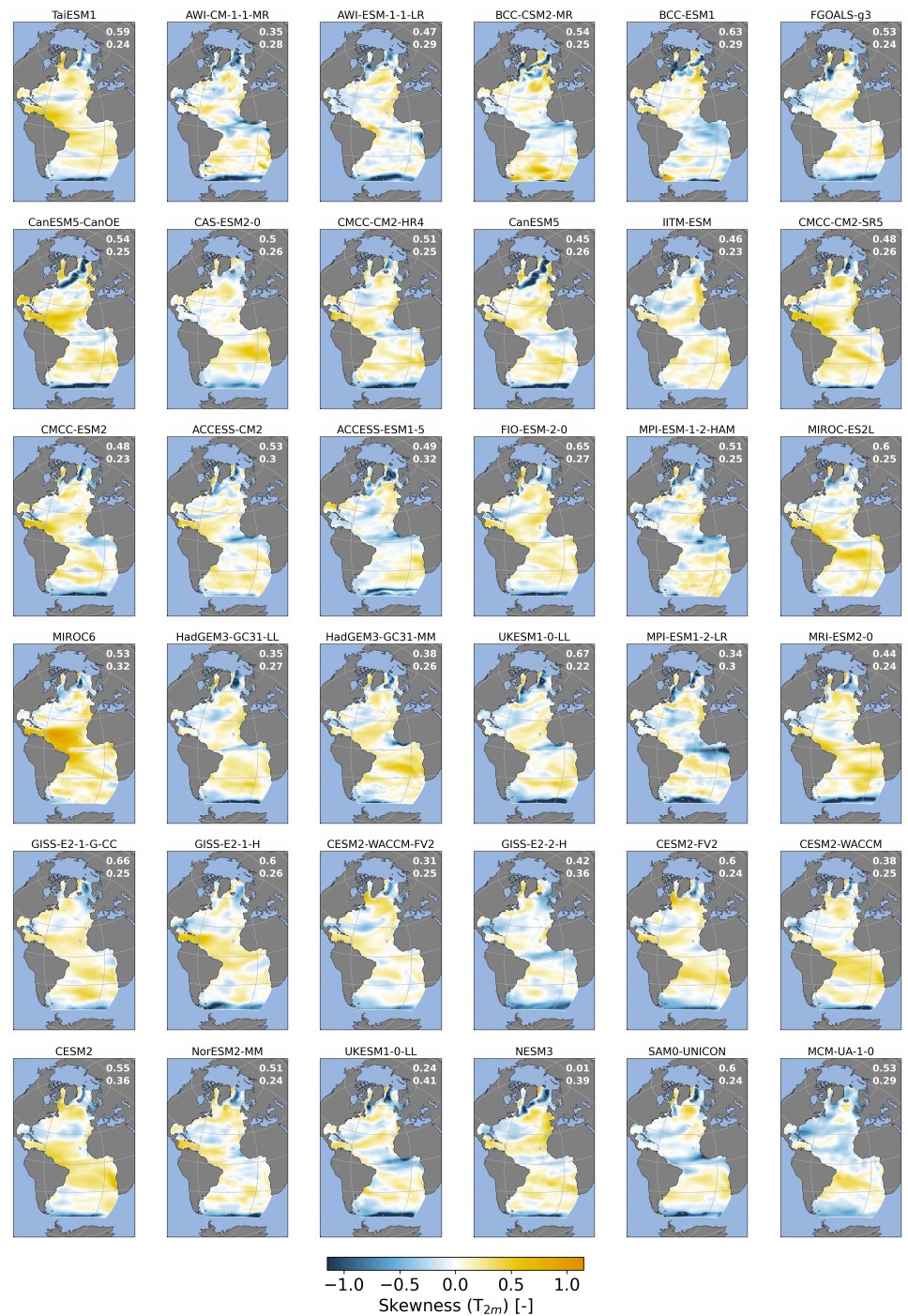

**Figure A14.** As Fig. A11 but for $T_{2m}$.

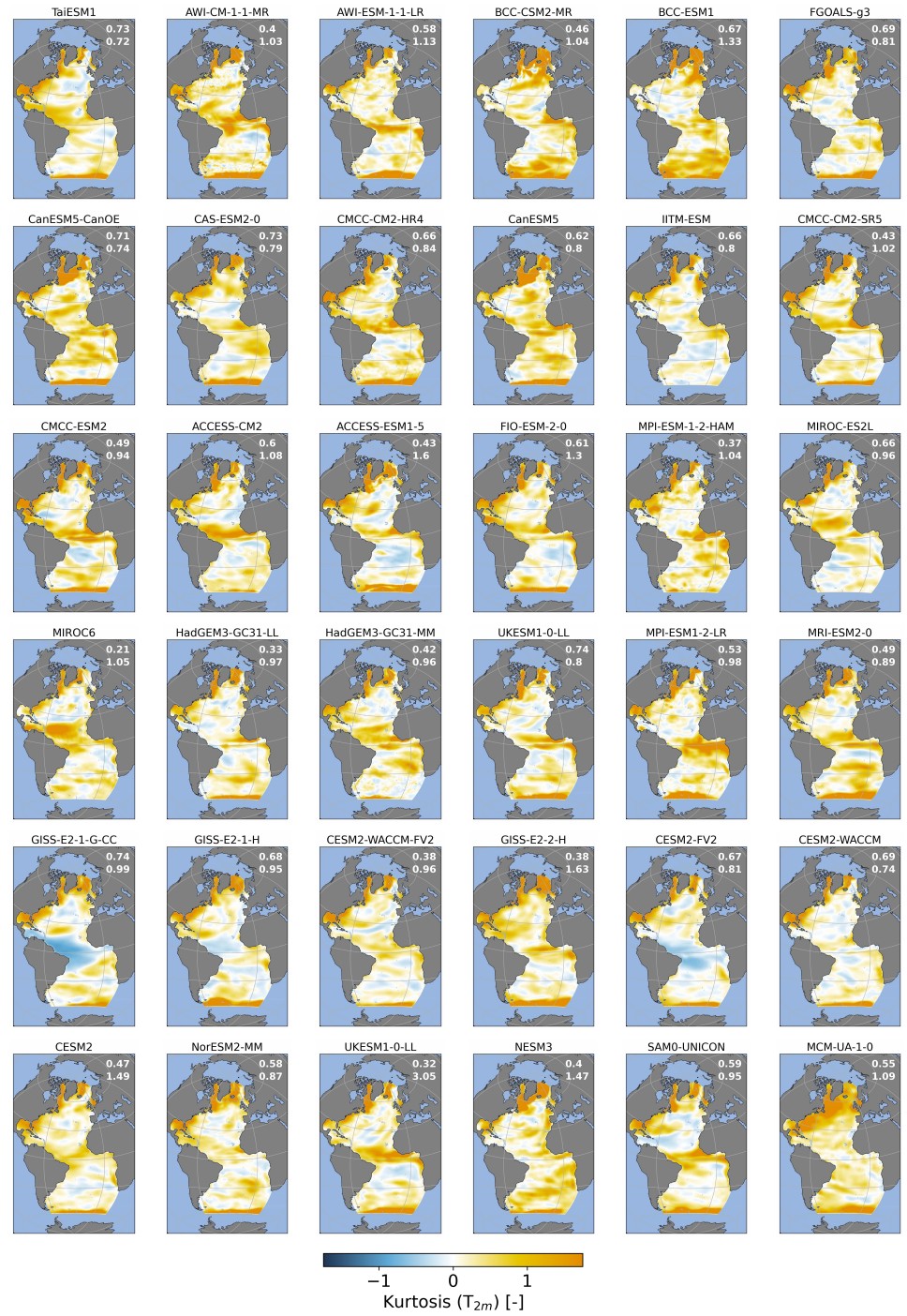

**Figure A15.** As Fig. A12 but for $T_{2m}$.

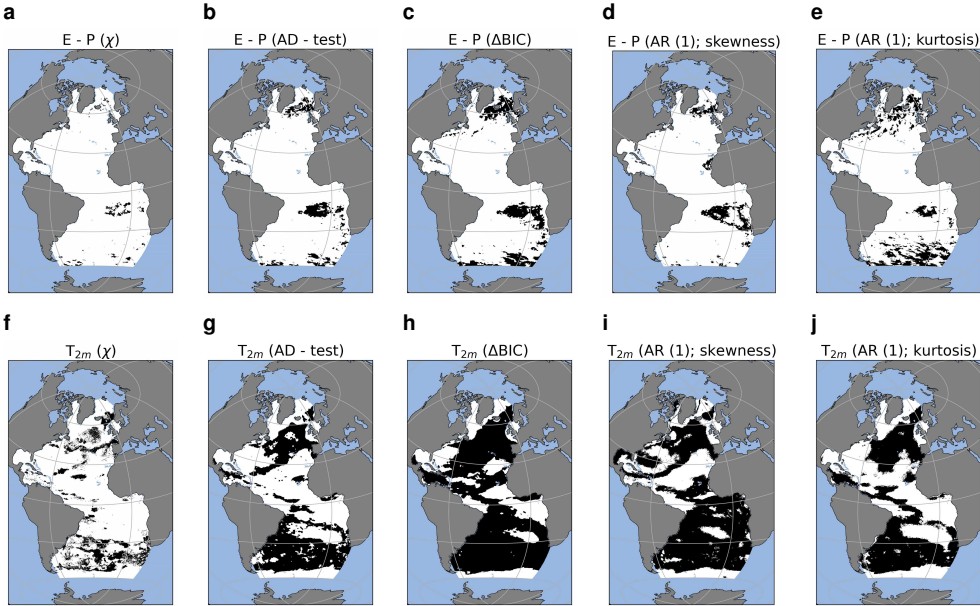

**Figure A16.** Collection of results for statistical tests. The top row is for the $E - P$ noise and the bottom row for the $T_{2m}$ noise. (a, e) $\chi$, where white regions represent grid points where the Normal Inverse Gaussian distribution provides a better fit than a Gaussian distribution. (b, f) Results from an Anderson-Darling test on normality where black regions represent grid points where the test is passed and the distribution is significantly Gaussian. (c, g) Results of the significance test of the skewness based on an AR(1) model where black regions represent that the AR(1) model provides a good fit. (d, h) as in (c, g) but for excess kurtosis.

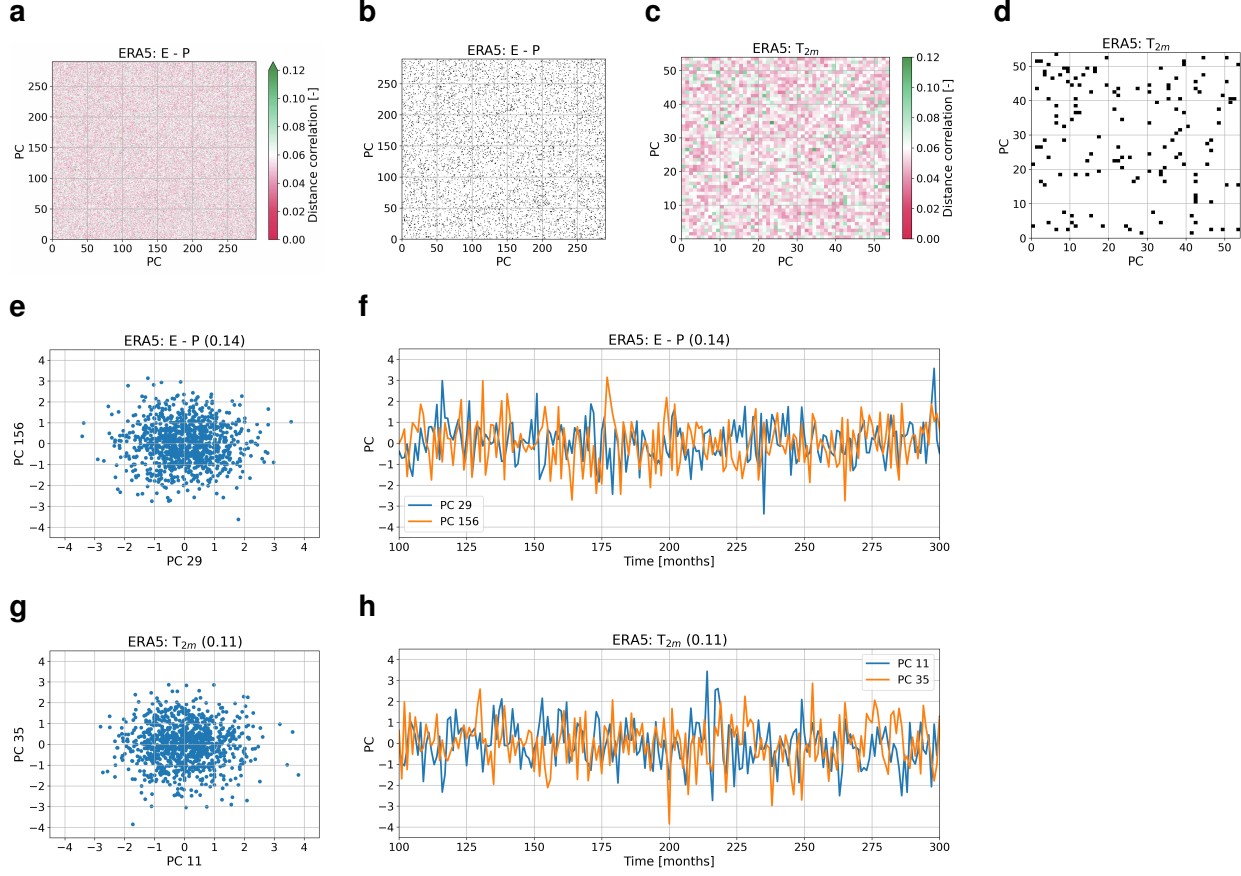

**Figure A17.** Results of the distance correlation analysis to test for dependencies between the PCs. (a) Distance correlation between the PCs for $E - P$. (b) Black squares represent a significant dependency ($p < 0.05$). (c) and (d) as (a) and (b) but for $T_{2m}$. (e) and (f) represent the two PCs that share the highest distance correlation (0.14) for $E - P$ with a scatter plot in (e) and a time series in (f). (g) and (h) are as (e) and (f) but for the PCs with the highest distance correlation (0.11) for $T_{2m}$. Note that a distance correlation has a range of 0 to 1, where 0 represents no relation, and 1 does represent a relation.

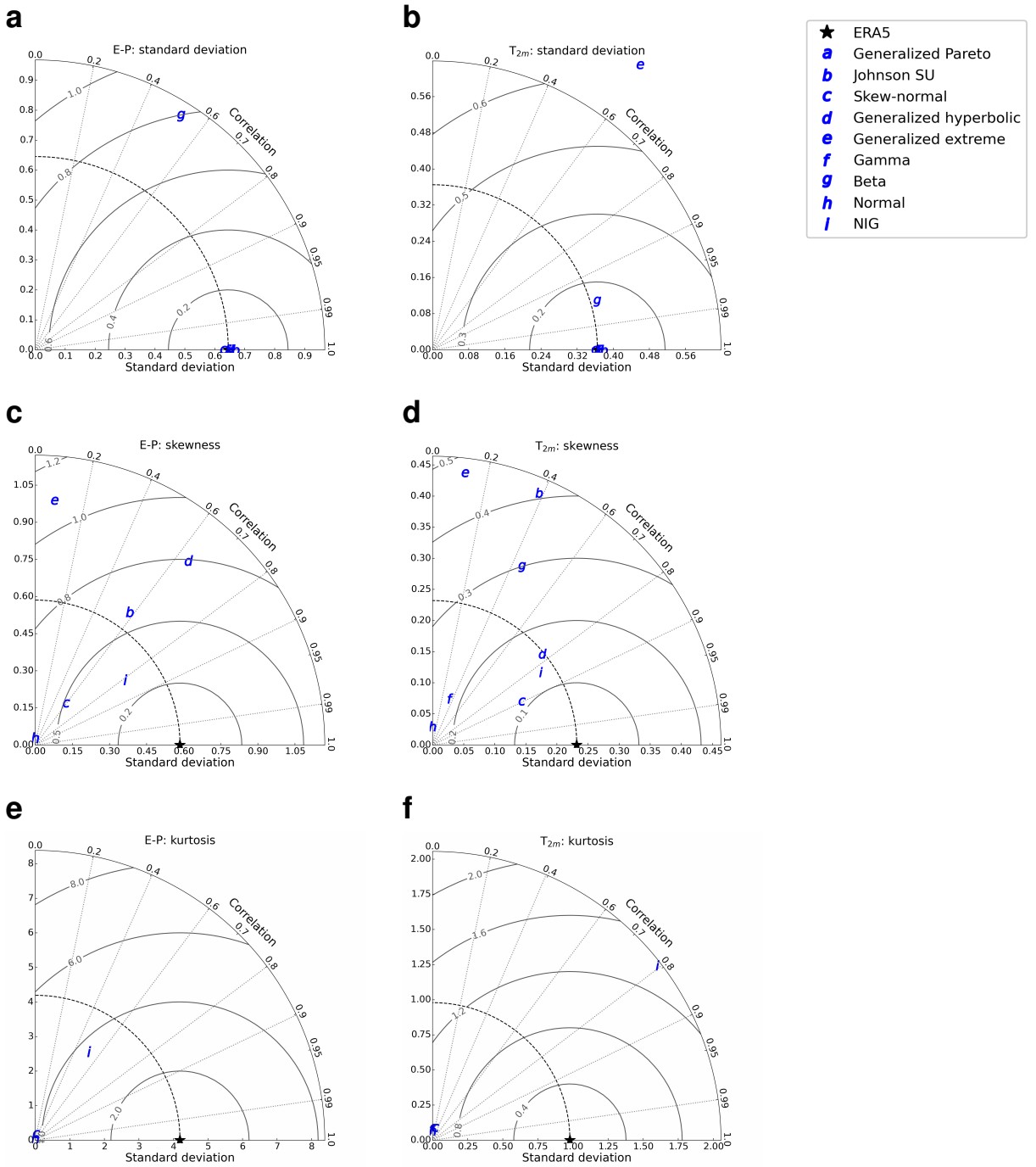

**Figure A18.** As Fig. 9 but for point wise fits using different distributions. Moments are based on 5000 realizations.

**Table A1.** CMIP6 model list.

| Number | Name | Reference |
| --- | --- | --- |
| 1. | TaiESM1 | Lee and Liang (2020) |
| 2. | AWI-CM-1-1-MR | Semmler et al. (2018) |
| 3. | AWI-ESM-1-1-LR | Danek et al. (2020) |
| 4. | BCC-CSM2-MR | Wu et al. (2018) |
| 5. | BCC-ESM1 | Zhang et al. (2018) |
| 6. | FGOALS-g3 | Li (2019) |
| 7. | CanESM5-CanOE | Swart et al. (2019a) |
| 8. | CAS-ESM2-0 | Chai (2020) |
| 9. | CMCC-CM2-HR4 | Scoccimarro et al. (2020) |
| 10. | CanESM5 | Swart et al. (2019b) |
| 11. | IITM-ESM | Choudhury et al. (2019) |
| 12. | CMCC-CM2-SR5 | Lovato and Peano (2020) |
| 13. | CMCC-ESM2 | Lovato et al. (2021) |
| 14. | ACCESS-CM2 | Dix et al. (2019) |
| 15. | ACCESS-ESM1-5 | Ziehn et al. (2019) |
| 16. | FIO-ESM-2-0 | Song et al. (2019) |
| 17. | MPI-ESM-1-2-HAM | Neubauer et al. (2019) |
| 18. | MIROC-ES2L | Hajima et al. (2019) |
| 19. | MIROC6 | Tatebe and Watanabe (2018) |
| 20. | HadGEM3-GC31-LL | Ridley et al. (2019a) |
| 21. | HadGEM3-GC31-MM | Ridley et al. (2019b) |
| 22. | UKESM1-0-LL (MOHC) | Tang et al. (2019) |
| 23. | MPI-ESM1-2-LR | Wieners et al. (2019) |
| 24. | MRI-ESM2-0 | Yukimoto et al. (2019) |
| 25. | GISS-E2-1-G-CC | NASA/GISS) (2019) |
| 26. | GISS-E2-1-H | NASA/GISS (2019a) |
| 27. | CESM2-WACCM-FV2 | Danabasoglu (2019b) |
| 28. | GISS-E2-2-H | NASA/GISS (2019b) |
| 29. | CESM2-FV2 | Danabasoglu (2019d) |
| 30. | CESM2-WACCM | Danabasoglu (2019c) |
| 31. | CESM2 | Danabasoglu (2019a) |
| 32. | NorESM2-MM | Bentsen et al. (2019) |
| 33. | UKESM1-0-LL (NIMS-KMA) | Byun (2020) |
| 34. | NESM3 | Cao and Wang (2019) |
| 35. | SAMO0-UNICON | Park and Shin (2019) |
| 36. | MCM-UA-1-0 | Stouffer (2019) |