# Peer review of "Observation based temperature and freshwater noise over the Atlantic Ocean"

_EGUsphere, 2024_

## Referee Comment (RC2)

Review of EGUsphere-2024-2431, "Observation based temperature and freshwater noise over the Atlantic Ocean" by A.A. Boot and H.A. Dijkstra.

Recommendation: Acceptable for publication after major revisions.

This study considers the statistics of monthly-mean 2m temperature and $E - P$ fluxes in ERA5 and CMIP6 Earth System Models. Focusing on the standard deviation, skewness, and reduced kurtosis the authors consider how well different noise models (both principal component based and pointwise) reproduce the observed statistics. It is found that the best fit to observed variations is obtained from local Normal Inverse Gaussian (NIV) distribution fits. Assessment of CMIP6 model performance demonstrated both systematic differences as well as substantial inter-model spread.

The study is interesting and the manuscript is clearly written. It is my assessment that a number of points require further elaboration and contextualization, and aspects of the presentation should be clarified. As such my recommendation is that the paper be accepted after major revisions. Detailed comments follow.

**Major Comments**

1. Throughout the manuscript, discussions of skewness and kurtosis are provided without accounting for sampling variability. While the record length of $O(10^3)$ is not short, sampling variability of skewness and particularly kurtosis will be nonzero. The fact that two independent simulations of the same ESM and the same length as the Reanalysis data generate substantially different higher-order statistics (LL 251-253) indicates that sampling variability may be appreciable. In order that the paper focus on robust non-Gaussian structures the revised manuscript should provide an estimate of the statistical significance of the skewness and reduced kurtosis relative. As a null hypothesis I recommend fitting the monthly $E - P$ and $T_{2m}$ data to Gaussian AR(1) processes and then generating sampling statistics from the fit model. This approach would account for the reduction in the effective number of statistical degrees of freedom in the assessment of statistical significance. Another approach that similarly accounts for serial dependence of the data would also be appropriate.

2. The non-Gaussianity of sea surface temperatures (admittedly not $T_{2m}$) was previously considered in Sura and Sardeshmukh (2008) https://doi.org/10.1175/2007JPO3761.1. I recommend relating the results of the present study to this earlier one. Sura and Sardeshmukh considered daily variability but it may be the case that for SST differences between daily and monthly variations are modest.

3. While the fields of statistical moments provide important spatial information about variability it has been my experience that valuable complementary information is provided by inspecting probability density functions at representative locations. As the fields of skewness and kurtosis generally show large-scale spatial variations it should be possible to find regions with representative pdfs. I recommend that the revised manuscript include figures showing such representative pdfs.

4. LL 138-139: I do not understand the description of the PC(1) method. My assumption is that it involves sampling the same time point from all PCs, but it is not clear if sampling is with or without replacement and in either case the number of potential samples is much greater than 996. In fact, reference is made later in the manuscript to generating realizations of 10000 members using this method. The revised manuscript should include a clear description of this method.

5. I completely agree with the interpretation that the PC(N) model fails to capture non-Gaussianity as it breaks any dependence that might exist between the PCs so PC-marginal non-Gaussianity will be suppressed by central limit theorem type reasoning. A complementary interpretation is that there *is* meaningful dependence between PCs which plays an essential role in generating the pointwise non-Gaussianity. In principle, a noise model could try to capture some of this dependence. The revised manuscript should include a discussion of accounting for dependence between PCs and how this could lead to a noise model with correct pointwise statistics but also allowing for spatial dependence (cf LL 284-286).

6. The Normal Inverse Gaussian distribution is one among many possible parametric models that could be used to fit the data (e.g. the skewed-t distribution). Why was the NIG distribution chosen for this particular application? Particularly given the fact that this parametric model cannot always well capture the kurtosis of the data (cf. Figs 6,7) the revised manuscript should justify the sole focus on this particular parametric distribution.

7. Section 5 is quite long, and much of the material repeats what was said in previous sections (eg. identifying where there are biases between the statistics of the data and of the MMM). I recommend revising this text to reduce its length and reduce overlaps with other parts of the manuscript.

**Minor Comments**

1. LL 49-50: ERA5 is based on observations but as the authors note as a reanalysis product it is a model simulation. I recommend avoiding use of "observations" in describing reanalysis products (particularly for a quantity like $E - P$ which is not assimilated).

2. Section 4: It is implied but not stated explicitly (that I noticed) that ERA5 data are used for the noise model. This point should be explicitly stated.

3. It appears that Figures 5 and 6 were swapped in the submitted manuscript. This should be corrected in the revised manuscript.

---

## Author Comment (AC1)

**MS-No.:** egusphere-2024-2431

**Title:** Observation based temperature and freshwater noise over the Atlantic Ocean

**Authors:** Amber A. Boot and Henk A. Dijkstra

**Point-by-point reply to reviewer #1**

**September 24, 2024**

We thank the reviewer for their careful reading and for the useful comments on the manuscript.

**Overview**

*The manuscript investigates the statistical characteristics of the noise in two variables affecting the Atlantic Meridional Overturning Circulation (AMOC): freshwater flux (E – P) and 2m air temperature ($T_{2m}$), obtained from the ERA5 reanalysis data from 1940–2022. The authors test the common assumption that the noise follows a Gaussian distribution, using three different models based on principal component analysis (PCA) and the negative inverse Gaussian (NIG) distribution and look at moments up to kurtosis. They find that the NIG model outperforms the others, except for excess kurtosis in sea ice covered regions in the $T_{2m}$ data. Analysis shows significant skewness and kurtosis in the data, and the authors conclude that the noise cannot be classified as white noise. In addition to the ERA5 reanalysis data, the authors also analyze 36 CMIP6 models and their multi-model mean (MMM), demonstrating that these models struggle to capture skewness and kurtosis and are outperformed by the NIG distribution for most metrics.*

*By analyzing the statistical properties of the noise, this paper addresses an important related to AMOC variability. Overall, the methodology appears sound, and the paper makes a meaningful contribution to the field. However, I have some comments that should be appropriately addressed by the authors before the manuscript is ready for publication:*

**Major comments:**

1. *The paper would benefit from having a more detailed discussion on*

*the statistical methodology, in particular relating to the Kolmogorov-Smirnov (K-S) test. From the code it seems that the conventional $\alpha$=05 significance level is used, but this should also be stated in the text for clarity and reproducibility. Moreover, considering the limitations of the Kolmogorov-Smirnov test with heavy-tailed distributions such as the NIG, the paper would benefit from considering alternative tests such as the Anderson-Darling test. Further discussion on the grid points that failed the K-S test would also be interesting.*

**Author's reply:**
We agree with the reviewer, and we thank the reviewer for pointing out the Anderson-Darling test.

**Changes in manuscript:**
We will perform the Anderson-Darling test as well and include a more thorough discussion on the results of these tests. This will include a discussion on the grid points that fail the tests.

2. *The paper employs a Normal Inverse Gaussian distribution (NIG) which presents a more flexible generalization of the normal distribution to allow for skewness and kurtosis to be expressed. Given that this model struggled to capture the excess kurtosis in certain areas it would be interesting to see the model compared with other models capable of expressing these additional moments, e.g. the generalized hyperbolic distribution or others.*

**Author's reply:**
We have tested several (more than 10) different distributions, among which the generalized hyperbolic distribution. None of these distributions performed better than the NIG distribution, which is the reason we chose the NIG distribution.

**Changes in manuscript:**
We will clarify why we chose the NIG model. Furthermore, we will, if applicable, include a discussion on whether other distributions are able to capture the kurtosis in regions where the NIG distribution performs relatively poorly.

3. *As the authors acknowledge, spatial coherence is lost when the models are fitted to each point individually. It would be beneficial for the authors to investigate or provide some discussion on how much this loss may affect the results.*

   **Author's reply:**
   We agree with the reviewer that such a discussion would improve the manuscript.

   **Changes in manuscript:**
   We will add a few sentences or a paragraph to the discussion on this issue.

**Minor comments:**

1. *I would suggest a more detailed explanation of the Taylor diagrams be included to make it more clear to readers unfamiliar with the concept. Some references would also be useful.*

   **Author's reply:**
   We agree.

   **Changes in manuscript:**
   We will provide a short introduction to the Taylor diagrams explaining the concept.

2. *I would like some more details on how the NIG model is fitted to each time series. Do the estimated parameters significantly deviate from those corresponding to an ordinary Gaussian distribution? A more detailed statistical analysis of the significance of these deviations could strengthen the argument that the noise is non-Gaussian.*

   **Author's reply:**
   We thank the reviewer for this suggestion. Such an analysis will indeed provide a stronger argument.

   **Changes in manuscript:**
   We will include more analysis, also based on comments from reviewer 2

and based on this comment, to strengthen the argument that the noise is non-Gaussian.

3. *I would like to see some more discussion on why the different PCA-based models were chosen.*

   **Author's reply:**
   We use three different PCA-based models. The PC(1) model is used to test whether the PCAs can in fact capture the statistics of the noise well. However, since this method is not fully stochastic (as explained in the paper) we also chose to use other models. The PC(N) model is in set-up very similar but more stochastic than the PC(1) method. As the PC(N) model also has a discrete number of values to sample from, we also used the PC(NIG) model, which does not have this problem.

   **Changes in manuscript:**
   We will provide a motivation why we use the different PCA-based models.

**Grammatical corrections:**
We thank the reviewer for pointing out these errors and we will follow all suggestions.

1. *Line 3 and 13: "noise-induce transitions" should be changed to "noise-induced transitions".*

2. *Line 6: I suggest changing ". . . shows best performance" to ". . . gives the best performance" or similar.*

3. *Line 20 and 296: "noise induced transitions" should be changed to "noise-induced transitions".*

4. *Line 22: I would suggest rewriting "Recently, also noise induced transitions have been studied in. . . " to "Recently, noise-induced transitions have also been studied in. . . ".*

5. *Line 60: "the negative of the summing of the variables" should be rewritten as "the negative sum of the variables".*

6. *Line 106: Change "deviates from 0" to "deviates from zero".*

7. *Line 109: "Multi model mean" should be changed to "Multi-model mean".*

8. *Line 127: "Special pattern" should be corrected to "spatial pattern".*

9. *Line 312: Fix the subscript formatting error.*

---

## Author Comment (AC2)

**MS-No.:** egusphere-2024-2431

**Title:** Observation based temperature and freshwater noise over the Atlantic Ocean

**Authors:** Amber A. Boot and Henk A. Dijkstra

**Point-by-point reply to reviewer #2**

**September 24, 2024**

We thank the reviewer for their careful reading and for the useful comments on the manuscript.

**Overview**

*This study considers the statistics of monthly-mean 2m temperature and E - P fluxes in ERA5 and CMIP6 Earth System Models. Focusing on the standard deviation, skewness, and reduced kurtosis the authors consider how well different noise models (both principal component based and pointwise) reproduce the observed statistics. It is found that the best fit to observed variations is obtained from local Normal Inverse Gaussian (NIV) distribution fits. Assessment of CMIP6 model performance demonstrated both systematic differences as well as substantial inter-model spread.*

*The study is interesting and the manuscript is clearly written. It is my assessment that a number of points require further elaboration and contextualization, and aspects of the presentation should be clarified. As such my recommendation is that the paper be accepted after major revisions. Detailed comments follow.*

**Major comments:**

1. *Throughout the manuscript, discussions of skewness and kurtosis are provided without accounting for sampling variability. While the record length of $O(10^3)$ is not short, sampling variability of skewness and particularly kurtosis will be nonzero. The fact that two independent simulations of the same ESM and the same length as the Reanalysis data generate substantially different higher order statistics (LL 251-253) indicates that sampling variability may be appreciable. In order that the*

*paper focus on robust non-Gaussian structures the revised manuscript should provide an estimate of the statistical significance of the skewness and reduced kurtosis relative. As a null hypothesis I recommend fitting the monthly E - P and $T_{2m}$ data to Gaussian AR(1) processes and then generating sampling statistics from the fit model. This approach would account for the reduction in the effective number of statistical degrees of freedom in the assessment of statistical significance. Another approach that similarly accounts for serial dependence of the data would also be appropriate.*

**Author's reply:**
We agree with the reviewer that the sampling variability may play a role here. We appreciate the suggestion of the test of statistical significance.

**Changes in manuscript:**
We will follow the suggestion of the reviewer and test the statistical significance.

2. *The non-Gaussianity of sea surface temperatures (admittedly not $T_{2m}$) was previously considered in Sura and Sardeshmukh (2008). I recommend relating the results of the present study to this earlier one. Sura and Sardeshmukh considered daily variability but it may be the case that for SST differences between daily and monthly variations are modest.*

**Author's reply:**
We thank the reviewer for pointing out the paper of Sura and Sardeshmukh. Even though there are definitely differences between SST and $T_{2m}$, and daily and monthly data, we think it is important to include this in the discussion to put our results in a more complete perspective.

**Changes in manuscript:**
In the discussion we will relate the results of Sura and Sardeshmukh to the results found in our paper.

3. *While the fields of statistical moments provide important spatial information about variability it has been my experience that valuable comple-*

*mentary information is provided by inspecting probability density functions at representative locations. As the fields of skewness and kurtosis generally show large-scale spatial variations it should be possible to find regions with representative pdfs. I recommend that the revised manuscript include figures showing such representative pdfs.*

**Author's reply:**
We agree with the reviewer that finding regions with representative PDFs would add value to the results.

**Changes in manuscript:**
Following the suggestion, we will look for representative PDFs and include them.

4. *LL 138-139: I do not understand the description of the PC(1) method. My assumption is that it involves sampling the same time point from all PCs, but it is not clear if sampling is with or without replacement and in either case the number of potential samples is much greater than 996. In fact, reference is made later in the manuscript to generating realizations of 10000 members using this method. The revised manuscript should include a clear description of this method.*

**Author's reply:**
All PCs are 996 months long. For the PC(1) method we uniformly sample one integer from 1 to 996. We apply this integer for all PCs. For example, if our integer is 7, then we sample the $7^{th}$ month of each PC to construct the noise model. Using this method we therefore have in total 996 different realizations to sample from. The sampling is performed using replacement. In the manuscript we have used a timeseries of 10,000 realizations, which means we sampled the integer 10,000 times to consequently construct the noise fields.

**Changes in manuscript:**
A clearer explanation of the method will be provided in the revised text.

5. *I completely agree with the interpretation that the PC(N) model fails to capture non-Gaussianity as it breaks any dependence that might exist*

*between the PCs so PC-marginal non-Gaussianity will be suppressed by central limit theorem type reasoning. A complementary interpretation is that there is meaningful dependence between PCs which plays an essential role in generating the pointwise non-Gaussianity. In principle, a noise model could try to capture some of this dependence. The revised manuscript should include a discussion of accounting for dependence between PCs and how this could lead to a noise model with correct pointwise statistics but also allowing for spatial dependence (cf LL 284-286).*

**Author's reply:**
We thank the reviewer for this complementary interpretation and the suggestion to discuss a possible noise model to capture some of the dependence between PCs.

**Changes in manuscript:**
We will add a discussion on this issue in the last section of the revised paper.

6. *The Normal Inverse Gaussian distribution is one among many possible parametric models that could be used to fit the data (e.g. the skewed-t distribution). Why was the NIG distribution chosen for this particular application? Particularly given the fact that this parametric model cannot always well capture the kurtosis of the data (cf. Figs 6,7) the revised manuscript should justify the sole focus on this particular parametric distribution.*

**Author's reply:**
We have tried several (more than 10) different distributions, among which the skewed-t distribution, but none of them performed better than the NIG distribution.

**Changes in manuscript:**
We will clarify that we chose the NIG distribution after testing several different statistical distributions because it performed best.

7. *Section 5 is quite long, and much of the material repeats what was said*

*in previous sections (e.g. identifying where there are biases between the statistics of the data and of the MMM). I recommend revising this text to reduce its length and reduce overlaps with other parts of the manuscript.*

**Author's reply:**
We agree.

**Changes in manuscript:**
Section 5 will be shortened following the suggestion of the reviewer.

**Minor comments:**

1. *LL 49-50: ERA5 is based on observations but as the authors note as a reanalysis product it is a model simulation. I recommend avoiding use of "observations" in describing reanalysis products (particularly for a quantity like E - P which is not assimilated).*

   **Author's reply:**
   We thank the reviewer for pointing this out.

   **Changes in manuscript:**
   We will use different wording for the ERA5 data.

2. *Section 4: It is implied but not stated explicitly (that I noticed) that ERA5 data are used for the noise model. This point should be explicitly stated.*

   **Author's reply:**
   The noise models are indeed based on ERA5 data.

   **Changes in manuscript:**
   We will explicitly state that the ERA5 data is used for the noise models.

3. *It appears that Figures 5 and 6 were swapped in the submitted manuscript. This should be corrected in the revised manuscript.*

**Author's reply:**
The reviewer is correct.

**Changes in manuscript:**
Figures 5 and 6 will be corrected.

---

## Referee Report (RR1)

Review of revised draft of EGUsphere-2024-2431, "Observation based temperature and freshwater noise over the Atlantic Ocean" by A.A. Boot and H.A. Dijkstra.

Recommendation: Acceptable for publication after minor revisions.

The authors have responded thoroughly to the concerns I raised in the original manuscript. I have a few minor comments on the revised draft; when these have been addressed it is my recommendation that the manuscript be accepted for publication.

1. K-means clustering is based on a distance metric. In the description of the clustering analysis, please indicate how (if at all) the standard deviation, skewness and kurtosis are standardized before the clustering is calculated, in order to avoid combining dimensionally inhomogeneous variables and dominance of the distance by variables with larger dynamical ranges.

2. L120: Please cite a reference describing the relationship between skewness and kurtosis for a system with multiplicative noise.

3. L217: I believe it is more appropriate to say that such grid points have variability that is "not statistically distinguishable from Gaussian", rather than saying that they "are likely Gaussian". The authors may consider revising the text accordingly.

4. L271-272: Please provide a reference to the distance correlation metric, as this is not commonly used in atmosphere/ocean science.

5. I am not surprised that the authors find weak dependence between the PC modes. Nevertheless, the fact that the original non-Gaussian fields can be reconstructed with the actual observed PC time series (not the sampled versions used in the surrogate models) indicates to me that such dependence must be present, even if it is subtle and difficult to model statistically. The sum in such reconstructions is also subject to the Central Limit Theorem. I recommend that a note to this effect be included in the revised manuscript.

---

## Author Response (AR2)

**MS-No.:** egusphere-2024-2431

**Title:** Observation based temperature and freshwater noise over the Atlantic Ocean

**Authors:** Amber A. Boot and Henk A. Dijkstra

**Point-by-point reply to reviewer #1**

November 22, 2024

We thank the reviewer for their careful reading and for the useful comments on the manuscript.

**Overview**

*The authors have successfully addressed my initial concerns with the manuscript and I do not have any strong objections to it in its current state, aside from a few corrections and suggestions which I will outline below.*

**Minor comments:**

1. *Comparison of distributions: The authors mention having tested more than 10 different distributions, where the NIG model performed the best. It would be interesting to know how these were compared, and I think it would benefit the paper to include an overview or a table summarizing their performance. If this comparison is outside the main focus of the paper, then perhaps it could find its way to the appendix.*

   **Author's reply:**
   We have used Taylor diagrams to assess the goodness of fit.

   **Changes in manuscript:**
   We have included an additional figure in the Appendix with the Taylor diagrams.

2. *Model comparison metrics: The authors have provided additional figures which show that the NIG model provides a significantly better fit compared to a Gaussian model for most clusters. However, this is somewhat expected, as the NIG model is more flexible. It would be useful to*

*include some metrics that assess fit while accounting for the number of model parameters, such as AIC or BIC.*

**Author's reply:**
We have determined the AIC and BIC values for the fits. For the $E - P$ noise, the NIG fit is a better fit compared to a Gaussian fit for 98% and 87% of the grid points for the AIC and BIC metrics, respectively. For the $T_{2m}$ noise this is 62% and 35%, respectively. This nicely supports the other tests carried out.

**Changes in manuscript:**
The additional results have been added, and two subpanels have been added to Figure A16.

3. *Manuscript length: The manuscript is somewhat lengthy and it would be beneficial to shorten it slightly, if possible.*

   **Author's reply:**
   We have considered this, but we believe that all the text is relevant to support and show the results.

   **Changes in manuscript:**
   No changes made.

**Specific comments**

1. *Figures A4 and A5 are not referred to or discussed in the main text.*

   **Author's reply:**
   Figures A4 and A5 are referred to in lines 110 - 111.

   **Changes in manuscript:**
   No changes necessary.

2. *Lines 205-211 regarding the Anderson-Darling test. I would suggest cleaning up the language a bit. The paragraph is a bit long and unclear.*

**Author's reply:**
Suggestion followed.

**Changes in manuscript:**
The text has been clarified.

3. *Line 213: "The model fails to provide good statistics" is somewhat vague. Please clarify what is meant by this.*

   **Author's reply:**
   What we mean is that the model does not capture the skewness and excess kurtosis well.

   **Changes in manuscript:**
   The text has been clarified.

4. *Line 163-164: The phrase "using replacement" should be changed to "with replacement".*

   **Author's reply:**
   Suggestion followed.

   **Changes in manuscript:**
   The text has been changed accordingly.

5. *Line 179: "is set-up very similar but more stochastic than..." should be changed to "is set up very similarly, but are more stochastic than...".*

   **Author's reply:**
   Suggestion followed.

   **Changes in manuscript:**
   The text has been changed accordingly.

6. *Line: 179: Please be more precise by what is meant by "More stochastic"?*

**Author's reply:**
What is meant with that is that the total number of possible realizations to sample from is larger.

**Changes in manuscript:**
We have reworded this sentence in the text.

7. *"timescale" and "time scale" used inconsistently.*

   **Author's reply:**
   We have made sure it is now used consistently.

   **Changes in manuscript:**
   Time scale has been changed to timescale.

8. *Line 203-204: Consider replacing "fail this test" with "do not pass", and reword "8 grid points fail (out of. . . )" to "8 grid points do not (out of. . . )".*

   **Author's reply:**
   Suggestion followed.

   **Changes in manuscript:**
   The text has been changed accordingly.

9. *Line 204: The nested parentheses could be simplified by dropping the inner set.*

   **Author's reply:**
   Suggestion followed.

   **Changes in manuscript:**
   The text has been changed accordingly.

10. *Line 205-206: The word "test" is used very frequently (four times) in a single sentence. I suggest a slight rewording to reduce repetition.*

**Author's reply:**
Suggestion followed.

**Changes in manuscript:**
The text has been rewritten.

11. *The NIG abbreviation is introduced in line 193, after the PC(NIG) model has been introduced. I would suggest introducing this earlier, for example in line 169 where the NIG distribution is first mentioned.*

    **Author's reply:**
    Suggestion followed.

    **Changes in manuscript:**
    The text has been changed accordingly.

**MS-No.:** egusphere-2024-2431

**Title:** Observation based temperature and freshwater noise over the Atlantic Ocean

**Authors:** Amber A. Boot and Henk A. Dijkstra

**Point-by-point reply to reviewer #2**

**November 22, 2024**

We thank the reviewer for their careful reading and for the useful comments on the manuscript.

**Overview**

*The authors have responded thoroughly to the concerns I raised in the original manuscript. I have a few minor comments on the revised draft; when these have been addressed it is my recommendation that the manuscript be accepted for publication.*

1. *K-means clustering is based on a distance metric. In the description of the clustering analysis, please indicate how (if at all) the standard deviation, skewness and kurtosis are standardized before the clustering is calculated, in order to avoid combining dimensionally inhomogeneous variables and dominance of the distance by variables with larger dynamical ranges.*

   **Author's reply:**
   The standard deviation, skewness and kurtosis were standardized before the analysis.

   **Changes in manuscript:**
   This has been clarified in the text.

2. *L120: Please cite a reference describing the relationship between skewness and kurtosis for a system with multiplicative noise.*

**Author's reply:**
Suggestion followed.

**Changes in manuscript:**
Sardeshmuhk and Sura (2009) has been added.

3. *L217: I believe it is more appropriate to say that such grid points have variability that is "not statistically distinguishable from Gaussian", rather than saying that they "are likely Gaussian". The authors may consider revising the text accordingly.*

   **Author's reply:**
   Suggestion followed.

   **Changes in manuscript:**
   The text has been changed accordingly.

4. *L271-272: Please provide a reference to the distance correlation metric, as this is not commonly used in atmosphere/ocean science.*

   **Author's reply:**
   Suggestion followed.

   **Changes in manuscript:**
   Szekely et al. (2007) has been added as a reference.

5. *I am not surprised that the authors find weak dependence between the PC modes. Nevertheless,the fact that the original non-Gaussian fields can be reconstructed with the actual observed PC time series (not the sampled versions used in the surrogate models) indicates to me that such dependence must be present, even if it is subtle and difficult to model statistically. The sum in such reconstructions is also subject to the Central Limit Theorem. I recommend that a note to this effect be included in the revised manuscript.*

   **Author's reply:**
   Suggestion followed.

**Changes in manuscript:**
A comment has been added in the discussion.